# Social transmission in the wild can reduce predation pressure on novel prey signals

Liisa Hämäläinen [1,2,3 ✉], William Hoppitt [4], Hannah M. Rowland [1,5,6], Johanna Mappes [3,7], Anthony J. Fulford[1], Sebastian Sosa[8] & Rose Thorogood[1,7,9]

Social transmission of information is taxonomically widespread and could have profound effects on the ecological and evolutionary dynamics of animal communities. Demonstrating this in the wild, however, has been challenging. Here we show by field experiment that social transmission among predators can shape how selection acts on prey defences. Using artificial prey and a novel approach in statistical analyses of social networks, we find that blue tit (*Cyanistes caeruleus*) and great tit (*Parus major*) predators learn about prey defences by watching others. This shifts population preferences rapidly to match changes in prey profitability, and reduces predation pressure from naïve predators. Our results may help resolve how costly prey defences are maintained despite influxes of naïve juvenile predators, and suggest that accounting for social transmission is essential if we are to understand coevolutionary processes.

[1] Department of Zoology, University of Cambridge, Cambridge, UK. [2] Department of Biological Sciences, Macquarie University, Sydney, NSW, Australia. [3] Department of Biological and Environmental Sciences, University of Jyväskylä, Jyväskylä, Finland. [4] School of Biological Sciences, Royal Holloway, University of London, Egham, UK. [5] Max Planck Institute for Chemical Ecology, Jena, Germany. [6] Institute of Zoology, Zoological Society of London, London, UK. [7] Research Programme in Organismal and Evolutionary Biology, Faculty of Biological and Environmental Sciences, University of Helsinki, Helsinki, Finland. [8] Université de Strasbourg, CNRS, IPHC, UMR 7178, Strasbourg, France. [9] HiLIFE Helsinki Institute of Life Science, University of Helsinki, Helsinki, Finland. ✉email: liisa.hamalainen@mq.edu.au

Traditional models of coevolution assume that interacting species exert selection on each other by influencing the fitness of alternative phenotypes[1], but rarely consider how behavioral interactions within species may drive coevolutionary dynamics. Work over the last decade has generated an explosion of interest in demonstrating how ecological change alters evolutionary change and vice versa, and how this can influence community dynamics and coevolution[2,3]. While this is a major advance, there is little research investigating how interactions *within* species drive eco-evolutionary feedbacks in coevolving systems. This is surprising, given increasing evidence of social transmission of information in a wide range of taxa[4]. Gene-culture coevolution models suggest that socially transmitted information could influence coevolutionary processes by altering selection pressures in the environment[5,6]. For example, if social information about enemies in the environment spreads rapidly in one species, it can intensify local defenses and select for counter-offenses in the interacting enemy species[7]. The effects for interacting species might be particularly important if social information is transmitted across generations, as it allows more rapid acquisition of behaviors than could be achieved through the spread of alleles[6]. Here we take advantage of a key paradigm in evolutionary biology, predators versus chemically defended prey, to test how social transmission alters predators' foraging behavior, and whether this rapidly changing predator environment can alter predation pressure on novel prey signals.

Prey have evolved many defense strategies to avoid predation, including aposematism, where individuals display their unprofitability (e.g., distastefulness or toxicity) with conspicuous warning signals[8,9]. However, conspicuous prey are an easy target for naïve predators who have yet to associate the warning signal with unprofitability. This creates a problem for the evolution of warning signals[10,11], and for the maintenance of aposematic prey that need to survive repeated outbreaks of naïve individuals in each predator generation[12]. The existence of undefended Batesian mimics that gain protection from predators by resembling aposematic model species makes the situation even more complex as mimics may weaken predator avoidance learning[13,14]. Traditionally, research on aposematism and mimicry has focused on predators learning by personal experience[9], and possible mechanisms explaining the evolution and maintenance of aposematism include dietary conservatism of predators[15], kin selection[16], and high attack survival of aposematic prey[17]. However, there is extensive literature showing that animals can acquire food preferences and aversions via social effects (reviewed in ref. [18]), and a number of studies with avian predators have now demonstrated that individuals also learn to avoid unprofitable food by observing the foraging events of others[19–25].

This social transmission has the potential to alter selection for prey defenses: social information about prey unprofitability might reduce predation on novel aposematic prey and therefore facilitate the evolution of aposematism[22–25], whereas social information about palatable mimics might increase predators' likelihood to sample both mimics and their defended models[24,26]. However, almost all previous studies have been conducted in captivity where predators face less complex foraging costs and fewer distractions than occur in nature and very few studies have investigated how educated predators use social information when food profitability changes and learned food preferences need to be reversed[24,27,28]. In the wild, social transmission of avoidance has been demonstrated only in a vervet monkey population where the majority of individuals were already trained to avoid unpalatable food[29]. How social information about food unprofitability spreads among naïve predators, therefore, remains untested in field conditions where individuals have opportunities to observe both conspecifics and heterospecifics[20,25] and learn from both

positive and negative feeding events of others (feeding on profitable/ unprofitable prey, respectively).

Here, we investigate how social information about defended prey and their palatable mimics spreads in a wild blue tit and great tit population. We use artificial prey, a well-established experimental method to test how predators learn about novel prey signals[10,23,30–32] and combine this with technological advances that now make it possible to identify individuals' foraging choices[33]. Our aim is to test (1) how quickly birds learn to discriminate novel palatable and unpalatable food and whether they use social information about positive and/or negative foraging experiences of others during this learning process, and (2) whether informed birds reverse their learned avoidance towards previously unpalatable food (defended 'models') and if this is influenced by the observation of others consuming similar but palatable food (edible 'mimics'). This represents a situation where predators learn to avoid novel aposematic prey and then encounter a population of palatable mimics that do not co-occur with their aposematic models. In our experiment, we used colored almond flakes as artificial prey items because these were novel to the birds. Almond flakes were offered to birds at three feeding stations. Although aposematic prey can be solitary, aggregated aposematic prey, such as hemipteran or lepidopteran larvae, have provided classic examples in studies of the evolution of aposematism[34]. The experiment was conducted at an established field site in Madingley Wood, Cambridgeshire during summer when juveniles were abundant and aposematic prey suffer from high predation[12]. Blue tits and great tits were fitted with RFID tags (approximately 89% of the birds visiting the feeders, see "Methods" section) and antennas at feeding stations enabled us to record each individual's visits. Based on these records, we constructed a social network of the bird population[35] (Fig. 1a) and then used this to estimate opportunities for learning from others when novel foods were presented that varied in profitability. Our results show that birds use social information to discriminate unpalatable and palatable food. This suggests that social transmission among predators can influence attack rates on defended prey and their mimics, and therefore shape the selection environment experienced by prey.

## Results

**Avoidance learning experiment.** We first investigated avoidance learning by offering birds differently colored palatable and unpalatable almond flakes using a paired-feeder design distributed across the study area (Fig. 1b). This was replicated with three different color pairs during the summer: red/green, blue/purple, and yellow/orange (unpalatable/palatable, Fig. 1c). Loggers at all of the feeders enabled us to record each individual's foraging choices between the color pairs. A total of 191 RFID tagged birds (blue tits: $n = 79$, great tits: $n = 112$) visited the feeders during the experiments (75% juveniles; Fig. 1a), and in each experiment (i.e., color pair) birds learned to discriminate palatable and unpalatable almonds within 8 days, by which time the proportion of visits to the unpalatable feeder decreased below 0.1 (Fig. 2a). This suggests that avoidance learning was similar in each experiment, although color pairs might have differed in their discriminability. Our main goal was to investigate social information use; therefore, as each color pair was distinguishable (see Supplementary Information), any visual differences among replicates should not influence our conclusions. There was no evidence of species-level differences in learning but we found that adults decreased their consumption of unpalatable almonds at a faster rate than juveniles (day$^2$ × age (juvenile): estimate = $-17.734 \pm 3.593$, $Z = -4.935$, $p < 0.0001$; day × age (juvenile): estimate = $5.296 \pm 3.853$, $Z = 1.374$, $p = 0.17$; Fig. 2a; Supplementary Table 1).

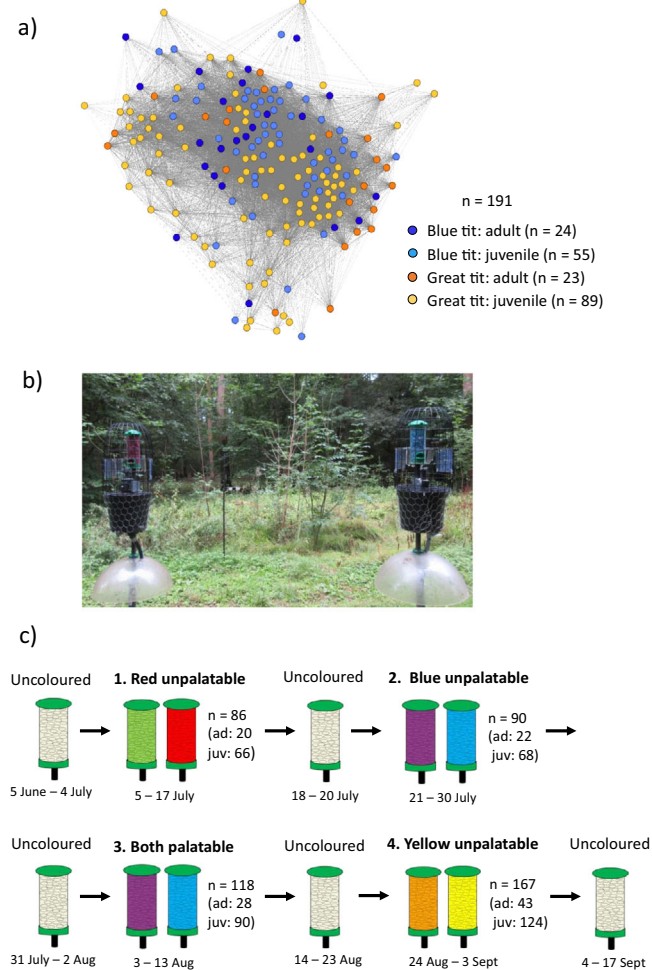

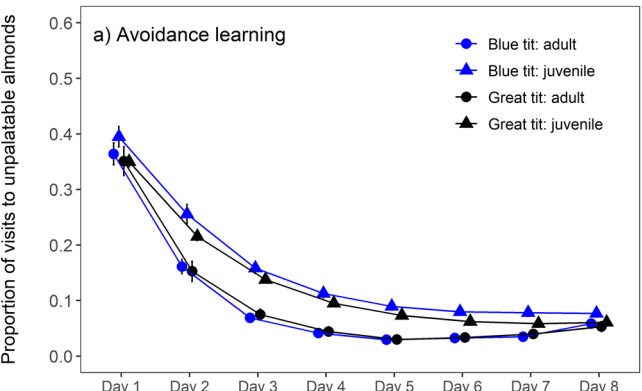

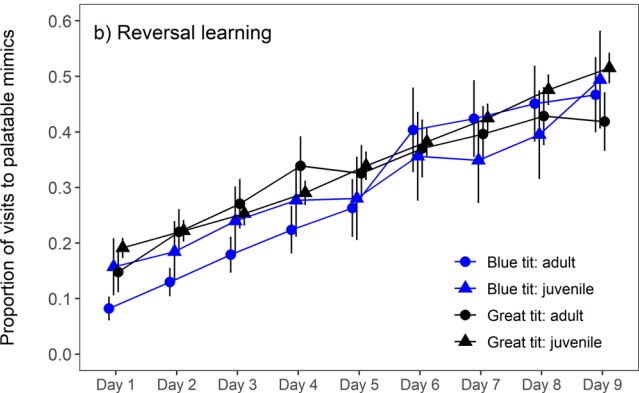

**Fig. 2 Learning across days.** The proportion of visits (**a**) to the unpalatable feeder in the avoidance learning experiments ($n = 189$ birds) and (**b**) to the feeder with palatable mimics in the reversal learning experiment ($n = 118$ birds). Graphs show the mean (±s.e.) proportion of visits across the days of the experiment (number of visits to each feeder divided by the total number of visits). All three avoidance learning experiments (red/green, blue/purple, and yellow/orange) are combined in the graph (**a**). Circles indicate the foraging choices of adults and triangles show the choices of juveniles (blue: blue tits, black: great tits). The plotted data were derived from the generalized linear mixed-effects models.

**Fig. 1 Summary of the experiments. a** The social network of the bird population was constructed based on the recorded associations outside the learning experiments. Nodes in the networks represent individual birds (adult blue tits = dark blue; juvenile blue tits = light blue; adult great tits = orange; juvenile great tits = yellow), and lines (edges) their associations in the network. **b** In each experiment, we had three feeding stations where birds were presented with a choice of two feeders with differently colored almonds. **c** Throughout the summer, we conducted three avoidance learning experiments with different color pairs (red/green, blue/purple, yellow/orange), and a reversal-learning experiment where both colors (blue/purple) were palatable. Between the learning experiments, birds were offered plain (palatable and uncoloured) almond flakes to collect information on their foraging associations.

We next investigated whether birds used social information during avoidance learning. If social avoidance learning occurs, we would expect birds to learn to avoid unpalatable food after observing the negative experience of close social affiliates. We devised a method to test this hypothesis by using social network analysis to estimate the number of times an individual was expected to have observed a social affiliate visiting the unpalatable feeder. The association network (Fig. 1a) was constructed based on social associations at the feeders outside the learning experiments (when birds were presented with uncoloured almonds; Fig. 1c), with the associations estimating the probability that two individuals were in the same group at a given time[35]. If birds learned socially to avoid the unpalatable food and/or prefer the palatable food by observing the choices of others, we expected avoidance of the unpalatable option (relative to the palatable option) to follow the connections of the association network,

since this should reflect opportunities for members of each dyad to observe one another. We, therefore, reasoned that the probability that the $i$th individual observed a specific feeding event by another individual, $j$, was proportional to the network connection between them, $a_{ij}$. In each avoidance learning experiment (i.e., color pair), we calculated the expected number of negative feeding events the individual had observed, prior to each choice as

$$O_{-,i}(t) = \sum_j N_{-,j}(t) a_{ij} \qquad (1)$$

where $N_{-,j}(t)$ was the number of times $j$ had visited unpalatable almonds prior to $i$'s choice at time $t$, and summation was across all birds in the network. The expected number of positive events was calculated in an analogous manner using the number of visits to palatable almonds. We then modeled each choice that the birds made (each visit to a feeder) and investigated how social information (expected number of observed positive and negative feeding events) and individuals' personal experience (previous visits to the palatable and unpalatable feeder) predicted their foraging decisions. To investigate whether a demonstrator's identity affected social information use, we further split the expected number of observed feeding events to observations of adults and juveniles, and to conspecifics and heterospecifics, and tested these effects separately. However, these social effects might

**Table 1 Summary of the asocial and social effects in three avoidance learning experiments.**

| | Estimate on log odds scale (SE) | | |
| | *Multiplicative effect on odds of choosing unpalatable colour [95% CI]* | | |
| | p-value (simulation p-value*) | | |
| | Mean number of actual feeds or mean expected number of observations (per bird) | | |
| | Red/Green | Blue/Purple | Yellow/Orange |
|---|---|---|---|
| **Positive personal experience** (per feed) | -0.106 (0.028)<br>x0.899 [0.851, 0.950]<br>p = 0.0001<br>mean = 43 feeds | -0.062 (0.021)<br>x0.940 [0.902,0.980]<br>p = 0.003<br>mean = 56 feeds | -0.028 (0.006)<br>x0.972 [0.961,0.983]<br>p < 0.0001<br>mean = 79 feeds |
| **Negative personal experience** (per feed) | -0.060 (0.027)<br>x0.942 [0.892, 0.994]<br>p = 0.028<br>mean = 8 feeds | -0.096 (0.017)<br>x0.909 [0.878,0.940]<br>p < 0.0001<br>mean = 9 feeds | -0.014 (0.004)<br>x0.986 [0.978,0.993]<br>p = 0.0002<br>mean = 12 feeds |
| **Observing a negative feeding experience** (per expected observation) | -0.080 (0.040)<br>x0.923 [0.853, 0.999]<br>p = 0.047 ($p_s$ = 0.004)<br>mean = 10 observations | -0.053 (0.025)<br>x0.948 [0.903,0.997]<br>p = 0.036 ($p_s$ = 0.002)<br>mean = 17 observations | -0.035 (0.006)<br>x0.965 [0.955, 0.976]<br>p < 0.0001 ($p_s$ = 0.0009)<br>mean = 41 observations |
| **Observing a positive feeding experience: conspecific** (per expected observation) | 0.000 (0.009)<br>x1.000 [0.983, 1.017]<br>p = 0.99 ($p_s$ = 0.17)<br>mean = 35 observations | -0.003 (0.005)<br>x0.997 [0.986, 1.007]<br>p = 0.51 ($p_s$ = 0.40)<br>mean = 64 observations | -0.004 (0.002)<br>x0.996 [0.993, 0.999]<br>p = 0.038 ($p_{s =}$ 0.92)<br>mean = 150 observations |
| **Observing a positive feeding experience: heterospecific** (per expected observation) | 0.015 (0.010)<br>x1.015 [0.996, 1.035]<br>p = 0.11 ($p_s$ = 0.028)<br>mean = 23 observations | 0.002 (0.006)<br>x1.002 [0.990, 1.015]<br>p = 0.70 ($p_s$ = 0.44)<br>mean = 35 observations | 0.007 (0.002)<br>x1.007 [1.004, 1.011]<br>p < 0.0001 ($p_s$ = 0.0009)<br>mean = 120 observations |

The probability of choosing the unpalatable option was analyzed using generalized linear mixed-effects models with a binomial error distribution. All observations of negative foraging experiences were pooled in the same social effect, but observations of positive foraging experiences were split between conspecifics and heterospecifics.
aThe simulation *p*-value ($p_s$) tests whether the putative social effect follows the social network as opposed to operating homogeneously among the birds. Significant asocial effects and social effects found to follow the network ($p$ < 0.05, $p_s$ < 0.05) are shaded gray.

be detected even if the effects were homogeneous within the population. This means that social learning may occur, but it does not follow the association network (i.e., all birds have the same probability of observing each other), which would provide weaker evidence of social transmission. Therefore, we further tested whether social effects estimated from our network differed from homogeneous effects, i.e., whether these effects followed our social network. A null distribution was generated by simulation assuming homogenous social learning and then compared to the social effects estimated from our network (see "Methods" section for full model specification and Supplementary Information for model validation).

The results of the avoidance learning models provided evidence of a social effect on birds' foraging choices, consistent with social learning resulting from observations of others consuming unpalatable almonds. After a greater number of expected observations of negative feeding events (as predicted by the network), individuals were less likely to choose the unpalatable color, and this effect was consistent across all three color pairs (Table 1). We also found that these effects followed our observed network (Table 1), which indicates the estimates from our network were a better predictor than estimates from a network where the observed effect was homogenized across all birds. In other words, this suggests that birds were more likely to learn from individuals that were closely associated with them in the social network, rather than having the same probability of learning from any individual, which strengthens our evidence of social transmission in the bird population. The effect of observing others consuming unpalatable almonds was similar whether birds observed conspecifics or heterospecifics, and we, therefore, pooled

| Social effect | Estimate per expected observation on log odds scale (SE) for transmission from juveniles or adults | | |
| | Wald test and p-value for difference between adults and juveniles | | |
| | Red/Green | Blue/Purple | Yellow/Orange |
|---|---|---|---|
| **Observing a negative feeding experience** | Juveniles: -0.086 (0.046) | Juveniles: -0.060 (0.038) | Juveniles: -0.006 (0.015) |
| | Adults: -0.774 (0.598) | Adults: -0.656 (0.439) | Adults: -0.381 (0.112) |
| | Z = 1.09, p = 0.28 | Z = 1.28, p = 0.20 | Z = 2.89, p = 0.004 |
| **Observing a positive feeding experience: conspecific** | Juveniles: 0.017 (0.012) | Juveniles: 0.003 (0.008) | Juveniles: 0.002 (0.003) |
| | Adults: -0.089 (0.055) | Adults: -0.114 (0.049) | Adults: -0.053 (0.013) |
| | Z = 1.84, p = 0.066 | Z = 2.13, p = 0.033 | Z = 3.58, p = 0.0003 |
| **Observing a positive feeding experience: heterospecific** | Juveniles: 0.030 (0.011) | Juveniles: 0.024 (0.010) | Juveniles: 0.018 (0.003) |
| | Adults: -0.090 (0.102) | Adults: 0.017 (0.060) | Adults: -0.051 (0.014) |
| | Z = 1.14, p = 0.25 | Z = 0.12, p = 0.90 | Z = 4.23, p < 0.0001 |
| | AIC difference of adding age specific transmission | | |
| | -8.69 | -12.81 | -97.57 |

**Table 2 Summary of differences in the effects of observing adults and juveniles in the three avoidance learning experiments.**

The probability of choosing the unpalatable option was analyzed using generalized linear mixed-effects models with a binomial error distribution.
aSignificant differences in social effects between adults and juveniles ($p < 0.05$) are shaded gray.

the observed negative feeding events in the final models (see "Methods" section and Supplementary Tables 3–5 for model selection procedure).

In addition to learning by observing the negative foraging experiences of others[19–21,23–25], predators might gather information about prey quality by observing avoidance behavior[22], and positive feeding events[27,36,37]. However, we found that the expected number of observed positive feeding events had a weaker and less consistent effect on birds' foraging choices compared to negative feeding events (Table 1). In the yellow/orange experiment, there was some evidence that an increasing number of expected observations of conspecifics eating palatable almonds made birds more likely to choose the same palatable color, but this effect did not follow the network. While the simulations to validate our modeling approach showed that we can reliably detect social avoidance learning, we found that the estimates of the effects of social appetitive learning were less conservative (see Supplementary Information). This suggests that we should not make strong conclusions about learning from positive feeding events unless we have good evidence that this effect follows the social network, and our result of social appetitive learning from conspecifics may therefore be an artifact of the analysis. Interestingly, expected observations of positive foraging events of heterospecifics made birds more likely to choose the opposite (unpalatable) feeder, although this effect was not significant in the red/green and blue/purple experiments (Table 1). This suggests that witnessing a strong response to unpalatable prey (e.g., vigorous beak wiping and head shaking) provides observers more salient social information than positive information about prey palatability, although this requires experimental tests that compare the effects of these two information types separately. Ignoring social information about prey unpalatability

might also be more costly because of the risk of consuming highly toxic prey. However, our experimental set-up with highly aggregated food items might have created competition at the feeders. This could provide another explanation for the inconsistent results of positive social information use[38,39] as birds might have chosen the more available feeder even after observing others feeding on palatable almonds of the opposite color.

In general, an expected number of observations of adults feeding on unpalatable or palatable almonds had a stronger effect on birds' foraging choices compared to an expected number of observations of juveniles (Table 2). This is in line with predictions that individuals often rely more on social information from older and more experienced individuals[40], however, this has rarely been demonstrated (but see ref. [41]). The difference was clearest in the yellow/orange experiment, with birds reducing their likelihood of choosing unpalatable (yellow) almonds when the expected number of observations of adults but not juveniles increased (Fig. 3, Table 2). Similarly, our estimate for the number of observations of adults consuming palatable (orange) almonds had a stronger effect on foraging choices compared to estimated observations of juveniles, and the same difference was found in the blue/purple experiment (Table 2). The coefficients for the expected number of observations of adults were larger than for the observations of juveniles also in the red/green experiment (Table 2), but this difference was not statistically significant (overlapping 95% confidence intervals), and we, therefore, cannot make strong conclusions about the relative age effects in the red/green experiment (see full models in Supplementary Tables 3–5). Overall, these results suggest that social information from adults facilitates rapid avoidance learning among juveniles, which could reduce the predation cost that aposematic prey faces when naïve predators are abundant[12]. However, in our experiment, all

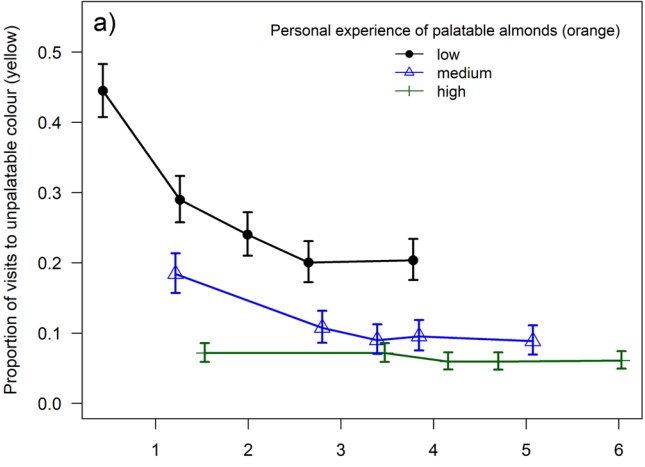

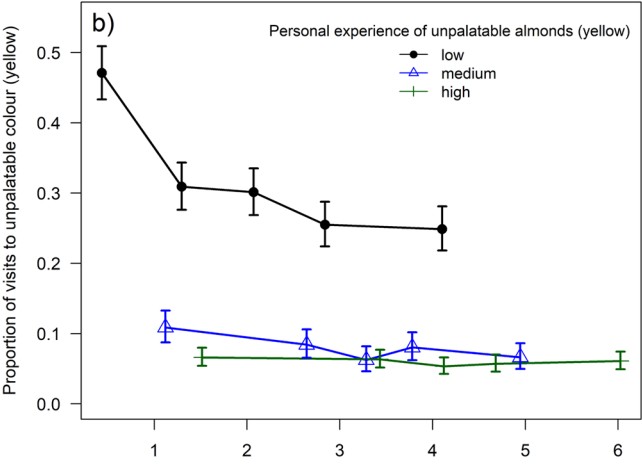

**Fig. 3 The effect of social information on birds' foraging choices.** The proportion of birds ($n = 168$) choosing the unpalatable (yellow) option in the yellow/orange experiment after observing adults feeding on unpalatable almonds. Social information reduced the likelihood to choose the unpalatable color when birds had little personal experience of (**a**) palatable or (**b**) unpalatable almonds (circles and black lines). For illustration purposes, the data is divided into 'personal experience categories' (represented by different symbols and colors) based on how many times birds had personally sampled palatable (**a**) or unpalatable (**b**) almonds before their current choice, standardized within each bird to allow us to show the within-bird patterns detected by the model. 'Low personal experience' includes data from birds within the 1st quartile for this variable, 'medium' birds within the 2nd quartile, and 'high' birds within the 3rd or 4th quartile. Within these 'personal experience categories', the data is further split into categories based on the expected number of observed unpalatable feeding events of adults. Symbols show the mean and 95% CI for the proportion of birds choosing the unpalatable option. See Supplementary Figs. 1 and 2 for the other two color pairs.

individuals were naïve to different food types, and juveniles, therefore, had opportunities to observe the negative feeding events of both adults and other juveniles. The opportunities to obtain social information by observing adults feeding on unpalatable prey might be reduced if adults are already informed, and it is possible that horizontal transmission of information from other inexperienced juveniles is more important when social information from adults is not available.

In our experiment, each food type was restricted to specific locations (feeders) simulating prey aggregation. The aggregation has been suggested to increase the survival of aposematic prey by enhancing avoidance learning and diluting the mortality cost if predators leave the aggregation after sampling one individual[10,32], as well as increasing initial wariness to attack warningly colored prey[42]. Our experiment suggests that aposematic prey might also benefit from aggregation by attracting the attention of many predators and increasing the likelihood that the negative foraging experience is witnessed by others. Nevertheless, previous studies with birds in captivity demonstrate that a single observation of others attacking aposematic prey can influence predators' foraging decisions[23–25], and social information may therefore be important also when prey are less aggregated and predators do not witness multiple predation events. In fact, the magnitude of social effects might be even higher than estimated in our experiment because our models assumed that all birds observed all feeding events in the groups in which they were present. In reality, birds were likely to observe only a proportion of these feeding events and the effect of one observation might therefore have a larger impact on birds' foraging choices than the coefficients in our models suggest.

**Reversal learning experiment.** While observing the negative feeding experiences of others may reduce predation on novel aposematic prey, social information about palatable mimics could in turn increase the predation risk of mimics and their defended models[26]. To investigate this, we next conducted a reversal-learning experiment with the blue/purple color pair by making both colors palatable after birds had acquired avoidance to blue almonds (Fig. 1c). We found that birds reversed their learned avoidance towards previously unpalatable (blue) almonds quickly: the proportion of visits to feeders with blue almonds increased already during the first day of the reversal learning experiment (compared to the last day of the avoidance learning experiment; Fig. 2) and in 9 days birds visited feeders with blue almonds as often as the alternative option (purple; Fig. 2b). Change in consumption across days (day$^2$ × age (juvenile): estimate = 5.496 ± 1.754, $Z = 3.133$, $p = 0.002$; day × age: estimate = 0.195 ± 1.928, $Z = 0.101$, $p = 0.92$) differed between adults and juveniles, with adults being more hesitant to sample blue almonds at the beginning of the experiment (Fig. 2b). Furthermore, blue tits tended to show greater hesitation to attack the previously unpalatable color compared to great tits but this difference was not statistically significant (day × species (great tit): estimate = −0.045 ± 0.025, $Z = −1.785$, $p = 0.07$; Supplementary Table 2).

If the birds were using social information during reversal learning, they would be more likely to start re-visiting the feeder containing the previously unpalatable blue almonds (mimics) after observing another bird feed there. Because our social network approach estimates opportunities for these observations, we expected the order in which birds first visited the blue feeder to follow the network connections, with birds being more likely to learn from individuals they were more closely associated with. To test this, we used network-based diffusion analysis (NBDA), which tests whether the order in which individuals acquire a behavioral trait follows a social network[43,44]. We found strong evidence that birds used social information in their decision to re-sample blue almonds for the first time, with the best-fit NBDA models including social transmission following our observed network (Table 3). We also found strong support for social transmission following our network when we compared Akaike weights of different social transmission models, with less than 1% support for the models with only asocial learning or social transmission following a homogeneous network. The

**Table 3 Summary of the best supported NBDA models of reversal learning with social transmission following the observed (models with ΔAIC < 2) or homogeneous network, or with asocial learning only.**

| Model: Equal/different asocial and social learning rates between the species | Transmission rate from adults/juveniles | Transmission rate from conspecifics/heterospecifics | AICc | ΔAICc | Social transmission parameter (s)[a] |
|---|---|---|---|---|---|
| **Social transmission following the observed network** | | | | | |
| Different asocial and social learning rates | different | same | 851.2 | 0 | 10.83 (adults) 0 (juveniles) |
| Equal asocial and different social learning rates | different | same | 852.2 | +1.0 | 23.06 (adults) 0 (juveniles) |
| **Social transmission following a homogeneous network** | | | | | |
| Equal asocial and social learning rates | same | different | 886.4 | +35.2 | 0.10 (conspecifics) 0.03 (heterospecifics) |
| **Only asocial learning** | | | | | |
| Equal asocial learning rate | NA | NA | 883.3 | +32.1 | constrained to 0 |

best-supported models (Akaike weight = 64%) included social transmission following our observed network with equal transmission rates from conspecifics and heterospecifics, but different rates from adults and juveniles (Table 3): an observation of adults feeding on blue almonds had a stronger effect on observers' decisions to sample the same color (estimated social transmission parameter from the best-fit model: 10.83, 95% CI: 1.60–67.26) than an observation of juveniles (estimate: 0, 95% CI: 0–1.75). This corresponds to an estimated 68% of birds learning from adults (95% CI: 52–80) and 0% (95% CI: 0–35) learning from juveniles, despite juveniles being more numerous in the population.

Our results indicate that palatable mimics lose protection rapidly when they do not co-occur with their defended models, and social transmission of information can further accelerate this reversal learning. This is in contrast to a previous experiment with great tits in captivity, in which birds did not reverse their learned avoidance after receiving social information about previously distasteful prey being palatable[24]. However, the perceived costs and benefits to attack palatable mimics were likely to differ between the two experiments, and the longer exposure to mimics in the field (compared to a short-scale experiment in captivity[24]) might have increased birds' willingness to sample mimics. In addition, attacks on palatable mimics can depend on the abundance of mimics[45] and alternative prey[46,47], and higher competition at the palatable feeder might have increased birds' motivation to re-visit the previously unpalatable feeder. Our finding of birds using both conspecific and heterospecific information also increases the opportunities to gather social information about palatable mimics, as some predator species are less risk-aversive than others[48] and therefore are more likely to sample previously unpalatable prey. This heterogeneity in social information could further accelerate predation on mimics, at least when their defended models are not present.

Although predator-prey interactions are one of the best-studied examples of coevolution[9], predators' avoidance or reversal learning at an individual level has never been tested in a population of wild predators. Our study suggests that naïve predators not only learn rapidly about prey profitability in nature but do so by observing the foraging behavior of others. Furthermore, we show that juveniles are more likely to use social information from adults than from other juveniles which supports theoretical work on social learning strategies[40]. Although we cannot determine the specific cues that birds used to discriminate different prey types, previous work has shown that birds use social information about prey signals[21–25], and we

also minimized opportunities to learn about prey location by switching the side of the feeders daily. Our findings are therefore consistent with previous experimental work with avian predators conducted under controlled conditions in captivity[19–25] and may help to resolve a long-standing puzzle about how conspicuous prey defenses can be maintained in the face of repeated influxes of naïve juveniles to hungry predator populations[12]. More importantly, our study indicates that social transmission across species boundaries can shape coevolutionary dynamics between antagonistic parties. Eco-evolutionary feedbacks occur when ecological conditions determine how selection can act at the genetic level, and when these changes also alter an individual or species' ecology[2,3]. In our study, social transmission rapidly altered the potential for selection to act on both chemically-defended prey and their mimics, and the social environment determined the ecological conditions under which predators foraged on different food types. How important social transmission is for predator-prey coevolution might, however, vary among predator and prey communities. For example, social information is likely to play a more important role when predators are social and forage in groups. Although this was the case in our study, further work is needed to quantify social interactions in other predator communities, especially in areas rich in aposematic prey species (e.g., tropical environments[26]). In our experiments, the prey were also aggregated, and investigating how predators use social information about more dispersed and solitary prey provides an interesting topic for future research. Although we focused here on predator-prey interactions, social transmission is likely to also influence eco-evolutionary dynamics in other coevolving systems. For example, in host and parasite communities, parasite transmission is influenced by patterns of social association, but hosts may also learn how to avoid infection by paying attention to the cues of others[7]. Or, plant pollinators such as honey bees learn about the pay-offs of novel foraging opportunities by observing both evolved communication systems (i.e., waggle dance[49]) and by observing the foraging behavior of other species[50]. Investigating how social environments influence the ecological context of coevolving species may therefore be necessary if we are to understand many complex coevolutionary processes in nature.

## Methods

**Study site.** The experiment was conducted at Madingley Wood, Cambridgeshire, UK (0∘3.2′E, 52∘12.9′N) during summer 2018. Madingley Wood is an established field site with an ongoing long-term study of the blue tit and great tit populations. During the autumn and winter birds are caught from feeding stations using mist nets and they are fitted with British Trust of Ornithology (BTO) ID rings. Since 2012, blue tits and great tits have been fitted with RFID tags (BTO Special Methods permit to HMR), which enables collecting data remotely about their foraging

behavior and social relationships. The study site has 90 nest boxes that are monitored annually during the breeding season. In 2018 chicks ($n = 325$) fledged successfully from 45 nest boxes (blue tits = 21, great tits = 24) and they were all ringed and fitted with RFID tags when they were approximately 10 days old. Because new juvenile flocks were arriving at our study site throughout the summer, we also conducted several mist-netting and ringing sessions in July and August to maintain a high proportion of blue tits and great tits ringed and RFID tagged for the experiments (on average 89%, see below). The study protocol was approved by the Animal Users Committee at the Department of Zoology, University of Cambridge.

**Food items**. We investigated birds' foraging choices by offering them colored almond flakes at bird feeders that were distributed throughout the wood. Before beginning the experiments, we allowed the birds to become familiar with the food items by providing plain 'control' almonds (plain and not colored) in paired feeders (1.5 m apart) at three locations (approximately 170 m from each other). The feeders were surrounded with metal cages to exclude larger birds, and we placed plastic buckets under the feeders to collect any spilled almonds and minimize birds' opportunities to forage from the ground. We introduced the feeders at the beginning of June when the nestlings had fledged and were beginning to forage independently, and continued to provide these plain almonds in between our learning experiments (Fig. 3c).

In the learning experiments, almond flakes were dyed with non-toxic food dye (Classikool Concentrated Droplet Food Colouring). We used three different color pairs: green (Leaf Green) and red (Bright Red), purple (Lavender Purple) and blue (Royal Blue), and orange (Satsuma Orange) and yellow (Dandelion Yellow). Almond flakes were dyed by soaking them for approximately 20 min in a solution of 900 ml of water and 30 ml of food dye and then left to air dry for 48 h. In the avoidance learning experiments, we made half of the almond flakes unpalatable by soaking them for one hour in 67% solution of chloroquine phosphate, following previously established methods from avoidance learning studies with birds in captivity[14,23–25]. The food dye was added to the solution during the last 20 min.

Red and green are common colors used by aposematic, or cryptic prey, respectively[9]. Therefore, we investigated whether blue tits and great tits had initial color biases towards these colors before starting the main experiment. Because we did not want the birds in our study population to have any experience of the colors before the main experiment, this pilot study was conducted in Newbury, which is 130 km from our main study site. Birds were simultaneously presented with two feeders containing red and green almonds (both palatable) for 30 min and the number of almonds of each color taken by blue tits or great tits was recorded using binoculars. The position of the feeders was switched after 15 min to control for any preferences for feeder location, and the test was repeated on 9 different days. We did not find any evidence that birds had initial color preferences ($t$-test: $t = 0$, df = 15.69, $p = 1$). For the other two learning experiments, we chose color pairs that were available as a food dye and as different from red and green in the visible spectrum as possible to avoid generalization across experiments. These color pairs (blue/purple and yellow/orange) had similar contrast ratios as green and red, based on their RGB values (measured from photographs, see Supplementary Information). Although avian and human vision is different, the discriminability of colors is likely to be similar[51], and rapid avoidance learning in each experiment shows that all colors were easily distinguishable. This was the main requirement for testing social information use, and subtle differences in color pair discriminability should only introduce noise to our data but not influence our conclusions.

**Learning experiments with colored almonds**. We conducted three avoidance learning experiments with different color pairs throughout the summer: red/green, blue/purple, and yellow/orange (unpalatable/palatable). In addition, we conducted a reversal-learning experiment with the blue/purple color pair by making both colors palatable after birds had acquired avoidance to blue almonds. Each experiment followed a similar protocol, in which birds were presented with colored almonds at the same three feeding stations where they were previously offered plain almonds. Each feeding station had two feeders, where one contained the palatable color and the other contained the unpalatable color (except in the reversal learning test when both colors were palatable). We switched the side of the feeders every day to make sure that birds learned to associate palatability with an almond color and not a feeder position. The feeders were filled at least once a day (or more often if necessary) to make sure that birds always had access to both colors. We continued each avoidance learning experiment until >90% of all recorded visits were to the feeder with palatable almonds, indicating that most birds in the population had learned to discriminate the colors. This took 7 days in the red/green experiment and 8 days in the other two color pairs (blue/purple and yellow/orange). The reversal learning experiment was finished after 9 days when 50% of the visits were to the previously unpalatable color (blue), indicating that most birds had reversed their learned avoidance towards it.

**Recording visits to feeders**. We monitored visits to all feeders using RFID antennas and data loggers (Francis Scientific Instruments, Ltd) that scanned birds' unique RFID tag codes when they landed on a perch attached to the feeder. During the learning experiments, each day we also recorded videos from all three feeding

stations (using Go Pro Hero Action Camera and Canon Legria HF R66 Camcorder). From the videos, we monitored the proportion of blue tits and great tits that did not have RFID tags and were therefore not recorded when visiting the feeders. We calculated the estimated RFID tag coverage for each day of the experiments by watching at least 100 visits to the feeders from the videos (divided equally among the three feeding stations) and recording whether blue tits and great tits had an RFID tag or not. We realized that the number of untagged individuals was very high (approximately 50% of all visiting birds) when we started the experiment with the first color pair (red/green; see Supplementary Fig. 3). We, therefore, stopped the experiment after two days and caught birds from the feeding stations with mist nets to fit RFID tags to new individuals. To maintain a high number of individuals RFID tagged for the other color pairs, we conducted a mist netting session a day before starting each experiment, as well as 4–5 days after it. We always switched the feeders back to containing plain almonds during mist-netting sessions to ensure that this would not interfere with the learning experiments. Apart from the first two days of the red/green experiment, the RFID tag coverage was on average 89% throughout the experiments (varying between 80 and 95%, Supplementary Fig. 3).

Birds were recorded every time that they visited the feeders, i.e., landed on the RFID antenna. However, it is possible that birds did not take the almond during every visit. To get an estimate of how often birds landed on the antenna without taking the almond, and whether this differed between palatable and unpalatable colors, we analyzed the visits to the feeders from the video recordings. We watched videos from the five first days of each experiment (i.e., different color pairs) and analyzed 60 visits to each color (divided approximately equally among the three feeding stations). We recorded whether the feeding event happened (birds ate the almond at the feeder or flew away with it) or whether birds left the feeder without sampling the almond. Because the number of visits to the unpalatable feeder was low during the last days of the avoidance learning experiments, we decided not to analyze avoidance learning videos after day five (but recorded visits from all days of the reversal learning experiment). We found that in avoidance learning experiments birds started to 'reject' unpalatable almonds after two days, i.e., they sometimes landed on the feeder but flew away without taking the almond (see Supplementary Fig. 4a). This change was not observed at palatable feeders where birds continued to consume almonds at a similar rate as at the beginning of the experiment (Supplementary Fig. 4a). In reversal learning, the proportion of visits that did not include a feeding event did not differ between purple and blue almonds: birds showed similar hesitation towards both colors at the beginning of the experiment, but this wariness decreased when the experiment progressed, with birds taking the almond during most of their visits (Supplementary Fig. 4b).

**Statistical analyses and model validation**

*Foraging choices in learning experiments*. We first analyzed how birds' foraging choices changed during the learning experiments using generalized linear mixed-effects models with a binomial error distribution. The number of times an individual visited each feeder on each day of the experiment was used as a bounded response variable, and this was explained by species (blue tit/great tit), individuals' age (juvenile/adult), and day of the experiment (continuous variable), as well as bird identity as a random effect. When analyzing avoidance learning, initial exploration of data suggested that results were similar across all three experiments, so we combined the experiments in the same model. To investigate whether learning curves differed between the species or age groups, the day of the experiment was included as a second-order polynomial term, and we started model selections with models that included a three-way interaction between species, age, and day[2]. Best-fitting models were selected based on Akaike's information criterion (see Supplementary Tables 1 and 2).

*Social network*. To investigate if birds used social information in their foraging choices, we first constructed a social network of the bird population based on their visits to feeders outside of the learning experiments, i.e., when birds were presented with plain almonds (in total 92 days, see Supplementary Information for the robustness of analysis to exclusion of network data before or after the experiment). We used only these data as individuals were likely to vary in their hesitation to visit novel colored almonds. We used a Gaussian mixture model to detect the clusters of visits ('gathering events') at the feeders[52] and then calculated association strengths between individuals based on how often they were observed in the same group (gambit of the group approach). These associations (network edges) were calculated using the simple ratio index, SRI[35].

$$\frac{x}{x + y_A + y_B + y_{AB}} \qquad (2)$$

where $x$ is the number of samples where individuals A and B co-occurred in the same group, $y_A$ is the number of samples where only individual A was seen, $y_B$ is the number of samples where only individual B was seen, and $y_{AB}$ is the number of samples where both A and B were observed in the same sample but not together. Network associations, therefore, estimated the probability that two individuals were in the same group at a given time, with the values scaled between 0 (never observed in the same group) and 1 (always observed in the same group).

*Social information use during avoidance learning: model description*. If social avoidance learning was occurring, then the more birds observed negative responses

of others feeding on the unpalatable feeder, the less likely they would be to choose the unpalatable feeder themselves. Thus, we expected the probability of bird $j$ choosing the unpalatable option at time $t$ to decrease with $R_{-,j}(t)$ (the real number of negative feeding events observed by $j$ prior to time $t$). Likewise, if appetitive social learning was occurring, then the more birds observed positive responses of others feeding on the palatable feeder, the more likely they would be so to choose the palatable feeder themselves (rather than the unpalatable feeder). So, we also expected the probability of $j$ choosing the unpalatable option at time $t$ to decrease as $R_{+,j}(t)$ (the real number of positive events observed by $j$ prior to time $t$) increased.

However, we could not test for an effect of $R_{-,j}(t)$ and $R_{+,j}(t)$ directly, since birds often ate the almond away from the feeder, and therefore the real number of observed feeding events could not be measured. Instead, we aimed to test for a pattern following the social network that is consistent with these social learning processes. We reasoned that the probability that one individual $i$, observes a specific feeding event by another individual $j$, was proportional to the network connection between them, $a_{ij}$ (probability they are in the same feeding group at a given time). Therefore, in each avoidance learning experiment (i.e., different color pair), we calculated the expected number of negative feeding events observed, prior to each choice (occurring at time $t$) as

$$O_{-,i}(t) = \sum_j N_{-,j}(t)a_{ij}, \qquad (3)$$

where $N_{-,j}(t)$ was the number of times $j$ had visited unpalatable almonds prior to time $t$ ($i \neq j$), and summation is across all birds in the network, and likewise for the expected number of positive feeding events:

$$O_{+,i}(t) = \sum_j N_{+,j}(t)a_{ij}, \qquad (4)$$

where $N_{+,j}(t)$ was the number of times $j$ had visited palatable almonds prior to time $t$ ($i \neq j$).

We analyzed whether the expected observations of positive and/or negative feeding events of others influenced the foraging choices in the avoidance learning experiments using generalized linear mixed-effects models with a binomial error distribution. We used each choice (i.e., visit a feeder) as a binary response variable (1 = unpalatable chosen, 0 = palatable chosen), with the probability that unpalatable feeder is chosen on feeding event $E$ given by $p_E = p_{-,i(E)}(t_E)$, where $i$ ($E$) is the individual that fed during event E and $t_E$ is the time at which event E occurred. We then modeled the probability of $i$ choosing the unpalatable option at time $t$ as:

$$p_{-,i}(t) = \text{logit}\left(\alpha + \beta_{\text{asoc}+}N_{+,i}(t) + \beta_{\text{asoc}-}N_{-,i}(t) + \beta_{\text{soc}+}O_{+,i}(t) + \beta_{\text{soc}-}O_{-,i}(t) + B_i\right), \qquad (5)$$

where $N_{+,i}(t)$ is the number of times a choosing individual had visited the palatable feeder (positive personal information), $N_{-,i}(t)$ is the number of times a choosing individual had visited the unpalatable feeder (negative personal information), $O_{+,i}(t)$ is the expected number of observed positive (positive social information) and $O_{-,i}(t)$ observed negative feeding events (negative social information). Bird identity was included as a random effect, $B_i$ (age and species were later added as variables, see below). Parameters $\beta_{\text{asoc}+}$ and $\beta_{\text{asoc}-}$ are the effects of asocial learning about the palatable and unpalatable foods, $\beta_{\text{soc}+}$ is the effect of social learning about the palatable food, and $\beta_{\text{soc}-}$ is the effect of social avoidance learning about the unpalatable food. Estimation of these parameters, with associated Wald tests and confidence intervals, allowed us to make inferences about which effects were operating and the size of these effects. To aid model fitting we standardized all predictor variables and then back-transformed the effects to the original scale (see Supplementary Tables 3–5 for the model outputs). To assess the importance of asocial and social effects, we also ran separate models that excluded either asocial or social parameters and compared them to the initial model in Eq. (5) using Akaike's information criterion (see Supplementary Table 6). However, in most cases, this reduced model fit significantly, and we, therefore, kept all parameters in the final models.

Our approach took $O_{-,i}(t)$ as a measure of $R_{-,j}(t)$, and $O_{+,i}(t)$ as a measure of $R_{+,j}(t)$-, which we termed the 'expected' number of observations of each type. Strictly speaking, $O_{-,i}(t)$ and $O_{+,i}(t)$ were upper limits on the expected number of observations, assuming that birds observed all feeding events in the groups in which they were present, whereas only an unknown proportion of such events ($p_o$) was observed. Therefore, the real expected number of negative/positive observations would be $E\left(R_{-,j}(t)\right) = p_o O_{-,i}(t)$ and $E\left(R_{+,j}(t)\right) = p_o O_{+,i}(t)$ respectively. Thus, the coefficient, $\beta_{\text{soc}-}$, for the effect of $O_{-,i}(t)$ could be interpreted as $\beta_{\text{soc}-} = p_o \beta'_{\text{soc}-}$ where $\beta'_{\text{soc}-}$ is the effect per observation. Note that since $p_o \leq 1$, and $\beta_{\text{soc}-} = \beta'_{\text{soc}-} p_o$, $\beta_{\text{soc}-}$ is more likely to underestimate than overestimate the effect per observation of a negative feeding event. An analogous argument applies to the coefficient, $\beta_{\text{soc}+}$, for the effect of $O_{+,i}(t)$.

*Social information use during avoidance learning: extension to test for species effects.* After fitting the initial model shown in Eq. (5), we further broke down the model to test whether individuals were more likely to learn socially by observing conspecifics

than heterospecifics. This was done by splitting the expected number of observed positive and negative feeding events to observations of conspecifics ($O_{+C,i}(t)$, $O_{-C,i}(t)$) and heterospecifics ($O_{+H,i}(t)$, $O_{-H,i}(t)$), and including these in the model as separate explanatory variables thus:

$$p_{-,i}(t) = \text{logit}\begin{pmatrix} \alpha + \beta_{\text{asoc}+}N_{+,i}(t) + \beta_{\text{asoc}-}N_{-,i}(t) \\ + \beta_{\text{soc},+H}O_{+H,i}(t) + \beta_{\text{soc},-H}O_{-H,i}(t) \\ + \beta_{\text{soc},+C}O_{+C,i}(t) + \beta_{\text{soc},-C}O_{-C,i}(t) \\ + B_i \end{pmatrix} \qquad (6a)$$

with $\beta_{\text{soc},-H}$ and $\beta_{\text{soc},-C}$ giving the effect of a negative observation of a heterospecific and conspecific, respectively, whereas $\beta_{\text{soc},+H}$ and $\beta_{\text{soc},+C}$ give the effect of positive observation of a heterospecific and conspecific, respectively. In general $-/+$ subscripts refer to negative/positive feeding events and C/H subscripts to feeding events by conspecifics/heterospecifics. By re-parameterizing the model thus:

$$p_{-,i}(t) = \text{logit}\begin{pmatrix} \alpha + \beta_{\text{asoc}+}N_{+,i}(t) + \beta_{\text{asoc}-}N_{-,i}(t) \\ + \beta_{\text{soc},H+}O_{+,i}(t) + \beta_{\text{soc},H-}O_{-,i}(t) \\ + \left(\beta_{\text{soc},C+} - \beta_{\text{soc},H+}\right)O_{+C,i}(t) + \left(\beta_{\text{soc},C-} - \beta_{\text{soc},H-}\right)O_{-C,i}(t) \\ + B_i \end{pmatrix} \qquad (6b)$$

we were able to test for a difference between observations of negative feeds by conspecifics and heterospecifics $\left(\beta_{\text{soc},C-} - \beta_{\text{soc},H-}\right)$ and between positive feeds by conspecifics and heterospecifics $\left(\beta_{\text{soc},C+} - \beta_{\text{soc},H+}\right)$.

For all experiments there was no evidence for a difference between $\beta_{\text{soc},-H}$ and $\beta_{\text{soc},-C}$ (yellow/orange: $Z = 0.803$, $p = 0.42$; red/green: $Z = 0.065$, $p = 0.95$; blue/purple: $Z = 1.113$, $p = 0.27$). However, there was some evidence of a difference between $\beta_{\text{soc},+H}$ and $\beta_{\text{soc},+C}$ in two of the three experiments (yellow/orange: $Z = 1.359$, $p = 0.17$; red/green: $Z = 1.417$, $p = 0.16$; blue/purple: $Z = 0.729$, $p = 0.47$). Consequently, we reduced the model down to:

$$p_{-,i}(t) = \text{logit}\begin{pmatrix} \alpha + \beta_{\text{asoc}+}N_{+,i}(t) + \beta_{\text{asoc}-}N_{-,i}(t) \\ + \beta_{\text{soc},-}O_{-,i}(t) + \beta_{\text{soc},+H}O_{+H,i}(t) + \beta_{\text{soc},+C}O_{+C,i}(t) \\ + B_i \end{pmatrix} \qquad (7)$$

for further analysis, i.e., with different effects for observations of conspecific/heterospecific positive feeds, but not of negative feeds. We did this for all color combinations (including blue/purple) to allow comparison across experiments (see Table 1). The R code used to run these models can be found in Supplementary data[53] in 'GLMM models Orange Yellow final.r'.

*Social information use during avoidance learning: simulations to test for a network effect.* Next, we tested whether the social effects we detected followed the social network. When using a network-based diffusion analysis (NBDA[43]), researchers can compare a network model with one in which the network has homogeneous connections among all individuals, but we found this to be unreliable for our model. Instead, we used a simulation approach to generate a null distribution for the null hypothesis of homogeneous social effects, taking the size of the social effects from the fitted models. We ran 1000 simulations (using the same procedure described above) for all social effects that were found to be significant in each avoidance learning model (each color pair; see Table 1). The total number of expected observations was kept equal, but we homogenized the observation effect across all birds by replacing the probability of bird $i$ observing a feed by bird $j$, previously $a_{ij}$, with $\sum_i a_{ij}/n$, where n is the number of birds in the experiment, (i.e., all birds had the same probability of observing each feeding event). The model was fitted to the simulated data each time to extract the $Z$ value (Wald test statistic) of the social effect of interest. The distribution of these values was then used as a null distribution to test whether our observed social effect differed from the effects that did not follow the social network. To this end, we calculated the proportion of simulations that yielded a $Z$ value as extreme or more extreme than that observed (judged by distance in either direction from the mean of the null distribution). The R code used to run these simulations can be found in Supplementary data[53] in 'Simulations to test if network effects follow network Orange Yellow.r'.

*Social information use during avoidance learning: extension to test for age effects.* We then aimed to test whether each of the three social effects detected differed based on the age class of the observed individual (adult versus juveniles). We, therefore, split the negative expected observations $O_{-,i}(t)$ into the expected observations of adults $O_{-A,i}(t)$ and juveniles $O_{-J,i}(t)$, each with its associated coefficient in the model $\beta_{\text{soc},-A}$ and $\beta_{\text{soc},-J}$. Likewise, we split positive observations of conspecifics as $O_{+CA,i}(t)$ and $O_{+CJ,i}(t)$ and positive observations of

heterospecifics as $O_{+HA,i}(t)$ and $O_{+HJ,i}(t)$ to give the model:

$$p_{-,i}(t) = \text{logit} \begin{pmatrix} \alpha + \beta_{\text{asoc}+}N_{+,i}(t) + \beta_{\text{asoc}-}N_{-,i}(t) \\ + \beta_{\text{soc},-A}O_{-A,i}(t) + \beta_{\text{soc},-J}O_{-J,i}(t) \\ + \beta_{\text{soc},+HA}O_{+HA,i}(t) + \beta_{\text{soc},+HJ}O_{+HJ,i}(t) \\ + \beta_{\text{soc},+CA}O_{+CA,i}(t) + \beta_{\text{soc},+CJ}O_{+CJ,i}(t) \\ + B_i \end{pmatrix} \quad (8a)$$

As before, –/+ subscripts refer to negative/positive feeding events, C/H subscripts to feeding events by conspecifics/heterospecifics, and A/J subscripts to feeding events by adults/juveniles. We also fitted a re-parameterized version allowing us to test for a difference between expected observations of adults and observations of juveniles for each of the three social effects:

$$p_{-,i}(t) = \text{logit} \begin{pmatrix} \alpha + \beta_{\text{asoc}+}N_{+,i}(t) + \beta_{\text{asoc}-}N_{-,i}(t) \\ + \beta_{\text{soc},-J}O_{-,i}(t) + \left(\beta_{\text{soc},-A} - \beta_{\text{soc},-J}\right)O_{-A,i}(t) \\ + \beta_{\text{soc},+H}O_{+H,i}(t) + \left(\beta_{\text{soc},+HA} - \beta_{\text{soc},+HJ}\right)O_{+JA,i}(t) \\ + \beta_{\text{soc},+C}O_{+C,i}(t) + \left(\beta_{\text{soc},+CA} - \beta_{\text{soc},+CJ}\right)O_{+CA,i}(t) \\ + B_i \end{pmatrix} \quad (8b)$$

The R code used to run these models can be found in Supplementary data[53] in 'GLMM models Orange Yellow final.r'. The main results of each model are presented in Table 2 and full model outputs in Supplementary Tables 3–5.

*Social information use during reversal learning.* To investigate social information use during reversal learning, we used the order of acquisition diffusion analysis (OADA), a variant of NBDA[43], which explores the order in which individuals acquire a behavioral trait[44]. The rate of social transmission between two individuals is assumed to be linearly proportional to their network connection, and the spread of trait acquisition is therefore predicted to follow the network patterns if individuals are using social information. We used NBDA to investigate whether the order of individuals' first visit to the previously unpalatable blue almonds (mimics) followed the network. We fitted several different models that included (i) only asocial learning, (ii) social transmission of information following a homogeneous network (equal associations among all individuals), or (iii) social transmission of information following our observed network. Models that included social transmission were further divided into models with equal or different transmission rates from adults and juveniles, and from conspecifics and heterospecifics, by constructing separate networks for each adult/juvenile and conspecific/heterospecific combination. To investigate whether asocial or social learning rates differed between blue tits and great tits, we included species as an individual-level variable. We then compared different social transmission models that assumed that species differed in both asocial and social learning rates, only in asocial or only in social learning rates, or that they did not differ in either (see Table 3). The best-supported model was selected using a model-averaging approach with Akaike's information criterion corrected for small sample sizes. All analyses were conducted with the software R.3.6.1[54], using *lme4*[55], *asnipe*[56], and *NBDA*[57] packages.

**Reporting summary.** Further information on research design is available in the Nature Research Reporting Summary linked to this article.

## Data availability

All data is available in Dryad (https://doi.org/10.5061/dryad.9s4mw6mcv). Source data are provided with this paper.

## Code availability

Codes for the analyses are available in Dryad (https://doi.org/10.5061/dryad.9s4mw6mcv).

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

## Acknowledgements

We thank Verity Bridger, Sheena Cotter, Victoria Franks, Juliet Franks, Malcom Franks, Cecilia Heyworth, and the Wicken Fen BTO Ringing Group for their help in the field, and Neeltje Boogert for discussions about the study. L.H. was funded by the Finnish Cultural Foundation and Emil Aaltonen Foundation and is currently supported by Jenny and Antti Wihuri Foundation. H.M.R. was supported by a research grant from the Royal Society (RG110122), an early career project grant from the British Ecological Society (ECPG 3569/4373), a research fellowship from the Institute of Zoology London, and is currently supported by the Max Planck Society. J.M. was supported by the Academy of Finland (#284666) and the University of Jyväskylä. R.T. was supported by an Independent Research Fellowship from the Natural Environment Research Council UK (NE/K00929X/1) and a start-up grant from the Helsinki Institute of Life Science (HiLIFE), University of Helsinki.

## Author contributions

L.H. and R.T. conceived the experiment and W.H., H.M.R., J.M. contributed to its design. H.M.R. and R.T. provided equipment and research materials. L.H. and A.F. conducted data collection and L.H., W.H., R.T., and S.S. performed analyses. L.H. led the writing of the manuscript. All authors contributed critically to the drafts and gave final approval for publication.

## Competing interests

The authors declare no competing interests.
