## [Peer Review File · Nature Communications]

REVIEWER COMMENTS

Reviewer #1 (Remarks to the Author):

I was very interested to read this study, however, I found there were numerous issues with clarity of the text and the methodology used.

A few of the major issues were:

--The labelling of visiting a feeder as predation – that is easy to fix. It's OK to motivate your study by the problem of aposematic prey, but it doesn't make sense to call the act of visiting a seed feeder as predation.

--Inconsistent terminology surrounding the “expected” measure of a bird observing another feeding. The text frequently slips into referring to these events as though they were directly measured, when they are not. What ground-truthing evidence do you have that this expected measure corresponds to actual cases of one bird having observed another?

--Related to this: the parameterization of the social networks. It looks like these were often based on social interactions taking place after a given experiment. But this ignores dynamics in the social network, and moreover, it ignores the fact that the experiments themselves may have influenced the networks structure. What is the evidence that this parameterization actually captures what a bird would have seen within the experiments?

--Why these colours were chosen. This experiment relies on visual colour learning. It is curious, then, that no information about the colour spectra of the food options and undyed almonds are provided. In some cases (yellow/orange), it even looks like colour learning did not take place.

There is an extensive literature from comparative and human psychology over the past 50+ years on the social learning of food preferences and avoidance. This research is overlooked when providing context for this work. See for example work by BG Galef, Galef and Giraldeau etc.

Main text

Lines 48-49: Suggest changing “ecological interactions within species” to “behavioural interactions within species” because that is a strength and focus here

Lines 49-51: I would remove the “however” part

Line 73: Rather than “individual learning”, suggest it is more accurate to call this “learning by personal experience” or direct experience. Social learning is still learning by an individual, the difference is that the learner doesn't personally experience the prey.

Line 93: “follow individual predators in the wild” this is misleading. In this study, you use passive tags, which do not allow us to “follow individuals in the wild” as in telemetry or GPS tracking. Rather, the passive tags allow individuals to be detected when visiting one or more specific location(s). Also, these tits are not predators in the context of your study, because you are looking at granivorous behaviour.

Line 95: Suggest replacing “educated” with “informed”

Lines 110-111: Should say: “a total of 191 tagged birds...”. Untagged birds are known to have visited the feeders too, so the 191 is limited to tagged birds.

I don’t see any information provided on the proportion of the Madingley populations that were tagged within the main text. Can you add this info? This is necessary for the reader to evaluate how well sampled the networks are for these analyses.

Line 113: It doesn’t make sense to call almond flakes “prey”. It should be “food item” or “food option” etc. And it shouldn’t be called “predation risk” (e.g., in figure 3)

Line 138: But this is a correlational analysis. Change “influenced their foraging decisions” to “predicted their foraging decisions” or “were associated with their foraging decisions” or similar.

Line 140, 141-142: Here you refer to “observed feeding events”, but you don’t know that individual j really did observe something, so this is misleading. Elsewhere you use “expected observed feeding events”. The switch in terminology is confusing and incorrect – we don’t really know what an individual had observed based on the measurements here.

Line 144: use of “observed social effects” ... the use of “observed” here is extra confusing because now you are talking about something different, you mean real data vs. null model (and not birds learning by having observed other birds). Use different terminology to make this distinction clear.

Line 142: unclear what is meant by: “observation effects followed our social network”. Explain to the reader what it means, operationally, to have a phenomenon “follow” a social network.

Line 151-152: same comment, it’s not clear what it means to have something “follow” a network

Line 160: should be “and” not “or”

Line 161: change “effects of observing” to “association with expected observed positive feeding events...” & same comment throughout

Line 176: same comment

Line 181: “only after seeing” same comment. This is presumed, but not actually measured, so the use of

strong language implying that a focal bird was known to have seen something is both misleading and confusing

Lines 233-235: same as above, explain what it means for something to “follow the social network”. If this is explained once, it would not have to be repeated, you could refer back to the initial explanation.

Line 266-268: It is misleading to say that this study provides experimental evidence that individuals learn about prey defences by observing the foraging behaviour of others, given that the ‘observation’ part is presumed, but never actually measured

Figures and Tables

Figure 1A: in this network diagram, the size difference between adult and juvenile nodes is too subtle to be useful; either make the size difference larger, or use another feature (e.g., light and dark blue for two different ages of blue tit; light and dark orange for great tit, etc)

Why are there so many great tit juveniles relative to adults? The ratio for blue tits is approx. 2:1 but for great tits it is nearly 4:1. What accounts for this difference?

How were the colour pairs chosen? Did you evaluate the discriminability of the colours within the feeder in the blue tit visual system? It looks like some of these colour pairs would be far less discriminable to the tit visual system than others, which makes it odd to compare these different replications without prior intention about the colour cues they create. Furthermore, in order for this protocol to be repeatable by others, the colour spectra for the food items (dyed and un-coloured) need to be provided.

Figure 2 – What do you mean by relative predation risk? Is it the probability of using one feeder vs. the other? The caption needs to define what the metric on the y-axis is. If it IS a probability, the axes should not go above 1.2 in Figure 2b; that wouldn’t make sense. Also, the two y-axes should be scaled the same. And, it should not be called predation risk, but a choice probability – a seed feeder does not get predated.

Figure 3 (and also Figs S2-S3) – The data in these figures don’t make sense. First, how could it be possible that the number of times a focal individual had fed (themselves) would be so much greater, even orders of magnitude greater, than the number of times they were expected to have observed another individual feed? Something doesn’t add up here, given that the number of other individuals feeding is so much greater than the focal bird, and that each bird has many social connections within the network.

Another issue is that the graph in Figure 3B indicates that focal individuals who had (themselves) visited the unpalatable feeder 9, 10 or even 20 times still visited that feeder again with up to 30% probability – this seems either implausible, or it suggests that the manipulation of palatability wasn’t actually unpalatable after all. Did color learning actually take place with the yellow-orange experiment? How could it be possible that a bird’s own experience of palatability would have no effect at all in Fig. 3B,

whereas only a few observations of another's experience could have such a strong effect? If nobody is gaining any personal learning, how could social learning occur? There couldn't be any social learning without personal learning.

Need to provide context for what the social transmission parameter means. Is it unbounded? What values would provide evidence of strong social transmission vs. weak social transmission, or no social transmission?

Methods

What was the schedule of the experiments shown in Figure 1 in time and space? (I include space here because it's not clear whether the experiment used more than one location for the feeder pairs). I can't find that information anywhere in the main text. Were the two feeders always provided in between experiments, with uncoloured almonds?

This schedule is also very relevant for making sense of the analyses done here. For example, it sounds to me like you composed a single network based on all co-location events that were during the non-experimental times to parameterize the (expected) probabilities of having observed things. But many of these occurred after particular experiments, and the social network would have varied through time, and may even been influenced by the experiment itself. Are the inferred observation events parameterized based on social interactions that occurred after a particular experiment? Given that the network would have been dynamic through time, why was that approach taken?

Lines 435: How many chicks? Sentence does not make sense

Line 439: What proportion? This information is needed to evaluate coverage.

Line 440: What Animal Care or ethics approval was obtained? It's not clear that Dept. Zoology constitutes approval by an Animal Care Committee.

Line 473: "were unlikely to be generalized to green or red, and that had similar contrast ratios as green and red, based on their RGB values (measured from photographs..." This is a poor approach for avian vision. From what I can tell, these experiments used colours that happened to be available in this particular line of food dyes. But did you check whether each pair of dyed almonds would be perceptually distinguishable? It seems odd that that was not evaluated for an experiment that relies on the birds' ability to learn visual associations with colours. After dyeing the almonds, how different are they in spectrometer measurements? How different are they as perceived through the material or opening of the feeder? Does this explain the heterogeneity in some of your results?

Line 536: "as a bound response variable" do you mean "bounded"?

Lines 540-541: You don't have enough levels for this to make sense as a random effect. It's not

reasonable to estimate variance from 3 options. It can only be a fixed effect; at least 5 or more levels are needed for a random effect.

Given that the two feeders (palatable, unpalatable) were swapped daily, the statistical analyses of feeder use should include time of day (or time after the switch). Early in day, the location is new.

Lines 554-555: There is not enough information here to understand how the networks were comprised. "the probability that two individuals were in the same group at a given time" isn't a complete explanation, because it would have a different value for the two nodes.

e.g., bird A is observed 10 times total and always in the presence of bird B,

but bird B is observed 100 times, including 10 times with A, and 90 times without A

Edge A in presence of B is 1, but edge B in presence of A is 0.1

How was a single edge defined for the purpose of different network-based analyses here?

Reviewer #2 (Remarks to the Author):

Review of "Social transmission alters the ecological dynamics of predators vs. prey"

This is a fascinating paper investigating the extent to which information about the palatability and unpalatability of novel prey can be gained by observing the behaviour and choices of others. Social learning is a hot topic, and the work is significant because it attempts to elucidate the nature of information exchange in the context of food choice within and between two species under field conditions. I do however have the following comments which I hope will be of use in preparing any revision. In particular I suspect that more can be done to quantify the relative roles of asocial (direct) and social (indirect) learning, and more could be done to elucidate the (potentially complex) statistical properties of the "cumulative experience" models that are fitted.

1. The cornerstone of the paper is the fitted logistic model (equation 3 line 591 and subsequent extensions) based on the expected (through a_{ij}) distribution of observed feeding events by others and the actual feeding experience of the focal individual. I'm very reassured to see that simulations were run to evaluate the type I error rate and bias associated with the assumed model structure not only because of the difference between real and expected observations, but also because (i) the cumulative predictors ("N") in consecutive decisions are not independent (they all tend to increase over successive time intervals, see 2 below) and (ii) a response for individual i at time $< t$ is also part of a predictor for individual j at $> t$ (there is feedback) In addition to simulation, I therefore wonder whether it would be appropriate and possible to apply classical model diagnostics to ensure some key assumptions are not violated, notably independence of residuals and logit linearity.

2. As noted above, the predictors are clearly multi-collinear: as time ticks by, all of the major component predictors will increase in size. I cannot find any note of this multi-collinearity (e.g. whether all other

predictors in the model are simultaneously or sequentially controlled for in the Wald tests) or awareness of its implications (high multi-collinearity can potentially obscure otherwise strong relationships).

3. I wondered about the mechanics of the binary choice and the role of time. Are consecutive choices of the same bird to a given feeder within a short time frame treated any differently from much more spread out (and clearly separate) visits (e.g. if green almonds are chosen three times in a row could it be because of spatial proximity, not learning). Also, it is ironic that seeing birds visit an unpalatable feeder serves to educate observers to avoid the same feeder. One might think choosing a particular coloured feeder would prompt others to follow rather than refrain (just as it does for palatable prey), which suggests that the observed disgust reaction is more important than the visit per se.

4. Why was the individual station (it appears there were three of them) not entered as a predictor in the model? After all, it is possible that the birds apply location-specific rules initially e.g. avoid red almonds in station A and only learn later that it generalizes across the other stations. Likewise, although the effect of the demonstrators' identity (species and age class) is explored at length, were there any consistent patterns in the focal individual's identity in their propensity to learn from others (id nested within age class within species)?

5. The effect sizes in Table 1 seem remarkably small (with multiplicative effects of 0.99 very close to 1 despite significance) but I had to remind myself this was per (cumulative) choice. What is really needed here is an estimate of the size of asocial (direct) learning parameters so we can judge how important direct experience is compared to indirect experience. After all, if the argument in the paper is that social learning "alters ecological dynamics" (title) then it would help to know how much of a difference it makes compared to asocial learning alone.

6. I had a hard time understanding the meaning of "relative predation risk for unpalatable prey". While I can guess, it needs a formal definition so we can understand why a value of 1 is consistent with no preference. Incidentally why does Fig 2A end at about 0.2 but it pick up again at 0.4 when the reversal learning experiment begins (perhaps some learning has already gone on in day 1)?

7. The surprising result to me in Figure 3 is not that social learning goes on (e.g. more potential observations of others controlling for an individual's direct experience reduces the probability of a bad choice) but that asocial learning is not apparent at all (all the coloured lines overlapped). You have to be very careful here however. Note that there may be something rather special, for example, about the birds that have not visited unpalatable almonds themselves and yet have (potentially) observed 4-5 other birds doing so. See McElreath Statistical Rethinking 2nd edition for a discussion of the "selection-distortion effect" (aka "collider bias").

8. The first experiment compared observed choice dynamics with models where the a_{ij} values were all identical (homogeneous). Why not also explore setting social learning coefficient to zero to help assess the importance of social learning (as when fitting the NBDA models in the reversal learning experiment)? More importantly, following the odd result outlined in (7) above, I think it would be

informative to see whether we lose any predictive power by dropping asocial (direct) learning entirely and see if social learning explains most of the variability in the probability of choosing unpalatable almonds.

9. The experiments are great, and while I understand where it is heading, I wonder whether the implications are a little overdone or at least premature. For example do we really know that (line 1) “social transmission alters ecological dynamics” and (line 33) “accounting for social transmission is necessary to understand coevolutionary processes”? Clearly social learning can reduce the burden of predation and therefore help (line 271) “resolve a long-standing puzzle”, but there are other excellent explanations for how aposematism spreads (e.g. kin selection, they survive encounters). The paper provides good field evidence of social learning about dietary preferences, but it is hard on the basis of this experiment to really know how crucial it is to understanding signal evolution.

My overall take is that this is an excellent study capitalizing on technological advances to monitor individual behaviour in well-established population. The paper is well written and argued and the results highlighting social learning in the context of diet choice are fascinating.

Minor stuff

Line 95 its not really “negative social information”, but information about something negative (unpalatability)

Line 113 no EVIDENCE OF species-level

Line 115 (and elsewhere) Perhaps better to say Day^2 (shorter and more explicit) than Day (polynomial)

Line 189 reducing their likelihood OF choosing

Line 491 why end when $> 50\%$ (while it can go over 50% by chance, the expectation is 50%)

Lines 581, you probably need a “j not equal to i” here and elsewhere

Lines 667 et seq. Why bold fonts for O?

Tom Sherratt

Reviewer #3 (Remarks to the Author):

This is a wonderfully written paper addressing an exciting topic. The authors have combined high-throughput tracking and cutting-edge social transmission analytical methods with an elegant experimental design that provides, in my mind, the most convincing demonstration yet of how social learning can promote the evolution of aggregated aposematic prey.

There is an inherent risk in studies such as this where inferences based on artificial paradigms (here, coloured almond flakes at mechanical bird feeders) are made about natural predator-prey interactions

in the wild. In this case, I feel that the implications for how aposematic prey may evolve are justified.

One minor 'issue' is that we cannot actually tell whether the birds are learning about the colouration of the individual almond flakes or the colouration of the feeders in this study, since these are perfectly correlated. Nevertheless, as the authors state on line 198 onwards, this is exactly the kind of natural scenario you find with many aggregating lepidopteran and hemipteran aposematic prey. The only change I feel is needed in an overall excellent paper is to therefore briefly mention somewhere that the study's implications about aposematism are indeed restricted to aggregating rather than solitary prey.

I find myself in the peculiar position of not being able to critique anything else about this paper, and I feel the authors should be congratulated for a wonderful piece of scholarship that has a good chance of becoming a classic in the field.

Reviewer #1 (Remarks to the Author):

I was very interested to read this study, however, I found there were numerous issues with clarity of the text and the methodology used.

A few of the major issues were:

--The labelling of visiting a feeder as predation – that is easy to fix. It's OK to motivate your study by the problem of aposematic prey, but it doesn't make sense to call the act of visiting a seed feeder as predation.

We have now re-worded this throughout the text, using "proportion of visits to the feeder" instead of "predation risk". We have also re-worded "prey" to "food items" or "almonds".

--Inconsistent terminology surrounding the "expected" measure of a bird observing another feeding. The text frequently slips into referring to these events as though they were directly measured, when they are not. What ground-truthing evidence do you have that this expected measure corresponds to actual cases of one bird having observed another?

Thank you for pointing this out! We have now made sure that we are consistent in using "expected number of observations". As mentioned in the methods (lines 526-528), birds often flew away with the almond and ate it away from the feeder, so we could not measure the absolute number of observations, which is often the case in field studies. Associations in the social network describe the probability that two birds were recorded foraging together, and based on this, we could get the best estimate of observations in the wild bird population. The core assumption of NBDA approach is that if there is social transmission, the probability to learn should be related to social network associations. This is the case in our study, which gives support to our measure of expected number of observations.

--Related to this: the parameterization of the social networks. It looks like these were often based on social interactions taking place after a given experiment. But this ignores dynamics in the social network, and moreover, it ignores the fact that the experiments themselves may have influenced the networks structure. What is the evidence that this parameterization actually captures what a bird would have seen within the experiments?

We agree that social networks are dynamic and can change through time. We recorded social interactions at the feeders outside learning experiments when birds were presented with uncoloured "control almonds" (see Figure 1c and lines 498-503). We then used all these records (in total 92 days) to construct a social network that was as accurate as possible. Of course, these type of field experiments always include some level of uncertainty and we do not have direct measures of what each bird observed. Nevertheless, our positive result of social effects following the network is consistent with the network at least approximating the learning opportunities. In fact, noise in the network should only make our estimated effects more conservative. Therefore, if we got a negative result, that might be because the network is not appropriate, whereas our positive result gives more confidence on the network measures.

In addition, we have now conducted extensive analyses to assess the robustness of our main yellow/orange experiment analysis to inclusion/exclusion of network data from different amounts of time before and after the experiment was conducted. We present this robustness analysis in the Supplementary Information (lines 203-249) and provide a summary here. We first found that taking data only from within the experiment (days when the experiment was halted for mist netting and ringing session and birds were presented with uncoloured almonds, see lines 456-460 in the main text) led to too much error as a result of reduction in sample size, so we rejected this as a possibility for the main analysis (though we still included it in the robustness analysis). We then found that in most cases the correlation with the full data network remained high, suggesting that in general, the network was changing slowly. However, there was a lower correlation when the data from after the experiment is dropped, supporting the reviewer's concern that the experiment may have influenced the network structure.

We therefore reanalysed the data using each of the 28 networks, and conclude (lines 243-249 in Supplementary Information):

“Overall, the analysis was very robust to inclusion/exclusion of data from before and after the experiment was conducted. The direction of effects is consistent as is the pattern of significance. The magnitude of effects is very similar in all cases except for the effect of conspecifics feeding on palatable almonds: here there is some variation in relative magnitude, but this effect is nonetheless always estimated as small and statistically non-significant. To avoid making an arbitrary decision on the amount of network data to include, we report the analysis based on the full data network in the main text.”

--Why these colours were chosen. This experiment relies on visual colour learning. It is curious, then, that no information about the colour spectra of the food options and undyed almonds are provided. In some cases (yellow/orange), it even looks like colour learning did not take place.

*We chose the first colour pair, red and green, based on our pilot tests that suggested that birds did not have biases towards them (methods lines 408-416). Since it was important to replicate the experiment with different colour pairs to get a better idea of the generality of the results but the choice of colours was limited, the rest of the colours were chosen based on what food dyes were available. Although it could be interesting to redo this experiment and test hypotheses about differences in discriminability of colours, the aim of our study was to test social information use in general. Therefore, we attempted to choose colours that were as different as possible in the visible spectrum (i.e. to our eyes), but did not measure their colour spectra because this was not relevant for our research questions. Although avian and human visions are different, a recent study found no difference in colour discrimination between chickens and humans (Olsson et al., 2015. Bird colour vision: behavioural thresholds reveal receptor noise. *J. Exp. Biol.* 218, 184-193), which suggests that the colours that looked different to our eyes were also different to bird eyes. We also found no evidence that the colours we chose affected the results: avoidance learning curves were similar for each colour pair (combined in Fig. 2a), with birds acquiring avoidance within 8 days (> 90% visits to the palatable feeder), suggesting that birds learned to discriminate all colour pairs quickly. We have now clarified this in the results (lines 123-126) and methods (lines 416-424).*

There is an extensive literature from comparative and human psychology over the past 50+ years on the social learning of food preferences and avoidance. This research is overlooked when providing context for this work. See for example work by BG Galef, Galef and Giraldeau etc.

Thank you for pointing this out. Since our paper focuses on social learning in predator-prey context and the space is limited, we were forced to leave out some of the extensive social learning literature. We have now added to the introduction a sentence that acknowledges this literature (lines 78-80) and cited a review paper from Galef & Giraldeau (2001).

Main text

Lines 48-49: Suggest changing “ecological interactions within species” to “behavioural interactions within species” because that is a strength and focus here

We have now changed this (line 51).

Lines 49-51: I would remove the “however” part

Done.

Line 73: Rather than “individual learning”, suggest it is more accurate to call this “learning by personal experience” or direct experience. Social learning is still learning by an individual, the difference is that the learner doesn’t personally experience the prey.

We have now changed this as suggested (line 76).

Line 93: “follow individual predators in the wild” this is misleading. In this study, you use passive tags, which do not allow us to “follow individuals in the wild” as in telemetry or GPS tracking. Rather, the passive tags allow individuals to be detected when visiting one or more specific location(s). Also, these tits are not predators in the context of your study, because you are looking at granivorous behaviour.

We have now changed this to “identify individuals’ foraging choices” (line 100).

Line 95: Suggest replacing “educated” with “informed”

Done (line 103).

Lines 110-111: Should say: “a total of 191 tagged birds...”. Untagged birds are known to have visited the feeders too, so the 191 is limited to tagged birds.

We have now added “RFID tagged birds” (line 119).

I don’t see any information provided on the proportion of the Madingley populations that were tagged within the main text. Can you add this info? This is necessary for the reader to evaluate how well sampled the networks are for these analyses.

Approximately 89% of the birds visiting the feeders were RFID tagged. We have now added this information in lines 107-108. More detailed description is included in the methods (lines 450-461).

Line 113: It doesn't make sense to call almond flakes "prey". It should be "food item" or "food option" etc. And it shouldn't be called "predation risk" (e.g., in figure 3)

We have now re-worded this throughout the text. We have also changed "predation risk" to "proportion of visits to the feeder" in the text and in Figures 2 and 3.

Line 138: But this is a correlational analysis. Change "influenced their foraging decisions" to "predicted their foraging decisions" or "were associated with their foraging decisions" or similar.

Done (lines 151-152).

Line 140, 141-142: Here you refer to "observed feeding events", but you don't know that individual j really did observe something, so this is misleading. Elsewhere you use "expected observed feeding events". The switch in terminology is confusing and incorrect – we don't really know what an individual had observed based on the measurements here.

We have now changed this to "expected number of observed feeding events" to be more consistent (line 153).

Line 144: use of "observed social effects" ... the use of "observed" here is extra confusing because now you are talking about something different, you mean real data vs. null model (and not birds learning by having observed other birds). Use different terminology to make this distinction clear.

We have now changed this to "social effects estimated from our network" (lines 159-160).

Line 142: unclear what is meant by: "observation effects followed our social network". Explain to the reader what it means, operationally, to have a phenomenon "follow" a social network.

This means that the social effects that were estimated from our network differed from homogeneous effects where all birds had the same opportunity to observe each other. We have now clarified this in lines 154-158.

Line 151-152: same comment, it's not clear what it means to have something "follow" a network

We have now added a sentence to clarify this: "In other words, this suggests that birds were more likely to learn from individuals that were closely associated with them in the social network" (lines 170-171).

Line 160: should be "and" not "or"
Corrected.

Line 161: change "effects of observing" to "association with expected observed positive feeding events..." & same comment throughout

We have now re-worded this to “expected number of observed positive feeding events” (lines 179-180) and used the same wording throughout the paragraph.

Line 176: same comment

We have re-worded this (lines 195-196) and used the same wording throughout the paragraph.

Line 181: “only after seeing” same comment. This is presumed, but not actually measured, so the use of strong language implying that a focal bird was known to have seen something is both misleading and confusing

We have re-worded this to “when the expected number of observations of adults but not juveniles increased” (lines 200-201).

Lines 233-235: same as above, explain what it means for something to “follow the social network”. If this is explained once, it would not have to be repeated, you could refer back to the initial explanation.

We have now explained this earlier in the text (lines 154-158) and reminded here that this means that “birds being more likely to learn from individuals they were more closely associated with” (lines 254-255).

Line 266-268: It is misleading to say that this study provides experimental evidence that individuals learn about prey defences by observing the foraging behaviour of others, given that the ‘observation’ part is presumed, but never actually measured

We have now re-worded this to “our study suggests” rather than “provides experimental evidence” (line 288).

Figures and Tables

Figure 1A: in this network diagram, the size difference between adult and juvenile nodes is too subtle to be useful; either make the size difference larger, or use another feature (e.g., light and dark blue for two different ages of blue tit; light and dark orange for great tit, etc)

We have now changed the colours as suggested.

Why are there so many great tit juveniles relative to adults? The ratio for blue tits is approx. 2:1 but for great tits it is nearly 4:1. What accounts for this difference?

This is an interesting question, but we can only speculate what caused this difference. We had approximately same number of blue tits and great tits nesting in our nest boxes (21 blue tits and 24 great tits, now added in line 379), so this does not explain the observed difference. It is possible that juvenile blue tits were dispersing to new areas more rapidly or were simply less likely to visit the feeders, but we do not have any data on this.

How were the colour pairs chosen? Did you evaluate the discriminability of the colours within the feeder in the blue tit visual system? It looks like some of these colour pairs would be far less

discriminable to the tit visual system than others, which makes it odd to compare these different replications without prior intention about the colour cues they create. Furthermore, in order for this protocol to be repeatable by others, the colour spectra for the food items (dyed and uncoloured) need to be provided.

As explained in response to the main comment, the colour pairs after the first red vs. green experiment were chosen based on food dye availability and visible spectrum, and the learning curves for each colour pair were very similar. We have now clarified this in lines 416-424. The protocol for colouring the almond flakes (dye brand and colour, amount of dye and length of soaking the almonds) is provided in the methods (lines 397-405), which makes the study repeatable.

Figure 2 – What do you mean by relative predation risk? Is it the probability of using one feeder vs. the other? The caption needs to define what the metric on the y-axis is. If it IS a probability, the axes should not go above 1.2 in Figure 2b; that wouldn't make sense. Also, the two y-axes should be scaled the same. And, it should not be called predation risk, but a choice probability – a seed feeder does not get predated.

Relative predation risk was calculated by dividing the number of visits to each feeder by the number expected by chance (i.e. 50% of the visits). It therefore gets a value of 1 when birds are visiting both feeders equally often. However, we agree that the probability of visiting the feeder is more straightforward and we have now used it instead of predation risk. We have also scaled the y axes to be the same in both figures and labelled them "Proportion of visits to unpalatable almonds/palatable mimics".

Figure 3 (and also Figs S1-S2) – The data in these figures don't make sense. First, how could it be possible that the number of times a focal individual had fed (themselves) would be so much greater, even orders of magnitude greater, than the number of times they were expected to have observed another individual feed? Something doesn't add up here, given that the number of other individuals feeding is so much greater than the focal bird, and that each bird has many social connections within the network.

We checked the x axis data from these figures and found they are correct, and certainly plausible. To illustrate: the mean number of yellow manipulations (across birds and events) is 11.0, whereas the mean number of yellow manipulations by adults is 6.6. The mean sum of connections to adult birds is 0.62, so we would expect the middle of the range for "Expected observations of adults feeding on unpalatable almonds" to be around $0.62 \times 6.6 = 4.1$, which is close to what we observed here. Bear in mind 75% of individuals in the network were juveniles. If we had plotted the expected observations of all birds, the mean total connection was 3.0 and the middle of the range would have been around 33.

Another issue is that the graph in Figure 3B indicates that focal individuals who had (themselves) visited the unpalatable feeder 9, 10 or even 20 times still visited that feeder again with up to 30% probability – this seems either implausible, or it suggests that the manipulation of palatability wasn't actually unpalatable after all. Did color learning actually take place with the yellow-orange

experiment? How could it be possible that a bird's own experience of palatability would have no effect at all in Fig. 3B, whereas only a few observations of another's experience could have such a strong effect? If nobody is gaining any personal learning, how could social learning occur? There couldn't be any social learning without personal learning.

*Thank you for pointing this out! There is strong evidence of asocial avoidance learning in the yellow/orange experiment (and blue/purple experiment, as well as reasonable evidence in the red/green experiment - we have added rows to Table 1 to show this). We checked why these patterns were not appearing in the figures, and found the main cause to be an error in the code that was jumbling the data and causing the lines to be plotted at the same height. However, we also realised that the plot is not quite showing what the model is detecting. Due to inclusion of random effects, the model picks up on patterns within birds, i.e. that birds became less likely to choose yellow the more times they chose it. Figure 3b shows the raw data, rather than the model effects, and we would not expect the effect to be necessarily manifested in differences in height among the lines. The main reason for this is variation among birds in their preference for orange. For example, some birds may tolerate the unpalatable taste more than others (e.g. Hämäläinen et al. 2020 *Predators' consumption of unpalatable prey does not vary as a function of bitter taste perception. Behav. Ecol.* 31, 383-391). Furthermore, birds may be forced to a suboptimal feeder if they are hungry and other birds are at the orange feeder, which could explain why birds did not completely abandon the yellow feeder. There is almost certainly going to be variation in this due to differences in dominance (across two species of different sizes). Such variation among birds is likely to offset any within-bird effect when looking at the raw data in the manner of Fig 3b, since birds that have a relatively high preference for yellow will also inevitably have a higher value for "previous visits to unpalatable almonds".*

This effect is illustrated clearly if we compare estimates from the GLMM in which we have strong evidence of a negative effect, with estimates from a GLM with no random effect and fit it to the same data:

1) with random effect

```
model_glmm_network<-glmer((Feeder=="Yellow")~RSactorOrangeManips +RSactorYellowManips
+ RSactorOrangeExpectedObs +RSactorYellowExpectedObs+(1|PIT), family="binomial",
data=newData)
```

```
summary(model_glmm_network)
```

```
#                Estimate Std. Error z value Pr(>|z|)
# RSactorYellowManips -0.22966  0.06648 -3.455 0.000551 ***
```

2) without random effect

```
model_glm_network2<-glm((Feeder=="Yellow")~RSactorOrangeManips +RSactorYellowManips +
RSactorOrangeExpectedObs +RSactorYellowExpectedObs, family="binomial", data=newData)
```

```
summary(model_glm_network2)
```

#	Estimate	Std. Error	z value	Pr(> z)
RSactorYellowManips	0.367982	0.036372	10.117	< 2e-16 ***

In the GLM we even have strong evidence of an effect in the opposite direction.

*Consequently, we have also now changed Figure 3 and Figures S1 and S2 in Supplementary Information so that they show within bird patterns. We standardized the actorOrangeManips (visits to palatable feeder) and actorYellowManips (visits to unpalatable feeder) variables **within each bird** and then split the data into lines based on the quantiles of these variables. Each line then represents the data from birds that are relatively earlier and later in the course of their own asocial experience of the orange (Fig 3a) or yellow (Fig 3b) almonds.*

Need to provide context for what the social transmission parameter means. Is it unbounded? What values would provide evidence of strong social transmission vs. weak social transmission, or no social transmission?

Social transmission parameter estimates the rate of transmission per unit connection in the social network. We have now included this under Table 3. Since this is dependent on the scale of the network, there is no general rule to equate specific values of s with strong or weak transmission (though $s=0$ would of course correspond to no social transmission). However, one can translate the effect into estimated % of birds learning by social transmission along each pathway which gives a good idea of the importance of social transmission. We have therefore added: "This corresponds to an estimated 68% of birds learning from adults (95% CI: 52-80) and 0% (95% CI: 0-35) learning from juveniles, despite juveniles being more numerous in the population" in lines 267-268.

Methods

What was the schedule of the experiments shown in Figure 1 in time and space? (I include space here because it's not clear whether the experiment used more than one location for the feeder pairs). I can't find that information anywhere in the main text. Were the two feeders always provided in between experiments, with uncoloured almonds?

We have now included the time schedule for the experiments in Fig 1c. We have also illustrated in the figure that uncoloured almonds were always provided between the experiments. In each experiment, we had three different locations for feeder pairs. This information is found in the methods (lines 388-390 and 430-432) and we have now also added it to the figure legend in Fig 1b (lines 321-322).

This schedule is also very relevant for making sense of the analyses done here. For example, it sounds to me like you composed a single network based on all co-location events that were during the non-experimental times to parameterize the (expected) probabilities of having observed things. But many of these occurred after particular experiments, and the social network would

have varied through time, and may even been influenced by the experiment itself. Are the inferred observation events parameterized based on social interactions that occurred after a particular experiment? Given that the network would have been dynamic through time, why was that approach taken?

See above our response about the parameterization of networks. The robustness analysis on inclusion/exclusion of network data is now included in Supplementary Information (lines 203-249).

Lines 435: How many chicks? Sentence does not make sense.

We have now re-worded this sentence (lines 378-380).

Line 439: What proportion? This information is needed to evaluate coverage.

We have now added the proportion in line 383. The RFID coverage is discussed in more detail in lines 450-461 (also see Fig. S3 in Supplementary Information).

Line 440: What Animal Care or ethics approval was obtained? It's not clear that Dept. Zoology constitutes approval by an Animal Care Committee.

Our research plan was reviewed by the Department of Zoology Animal Users Committee (now added in lines 383-384) which performs Ethical Review for experiments involving animals that are not subject to legislation under the Animal Scientific Procedures Act (ASPA). Because birds were free to choose the food they consumed, the use of animals was not-regulated by ASPA. No other permits were required, after consultation with Natural England. The Animal Users Committee consists of the Deputy Director of Operations & Facilities of the University Biomedical Services, the Named Animal Care and Welfare Officer (NACWO), University Veterinary Surgeon, the Head of Department of Zoology, Chairperson of the Department Management Committee, and two academic members of staff.

Line 473: "were unlikely to be generalized to green or red, and that had similar contrast ratios as green and red, based on their RGB values (measured from photographs..." This is a poor approach for avian vision. From what I can tell, these experiments used colours that happened to be available in this particular line of food dyes. But did you check whether each pair of dyed almonds would be perceptually distinguishable? It seems odd that that was not evaluated for an experiment that relies on the birds' ability to learn visual associations with colours. After dying the almonds, how different are they in spectrometer measurements? How different are they as perceived through the material or opening of the feeder? Does this explain the heterogeneity in some of your results?

Please see above our answer to the comment on the colour pairs. Our results show that in each colour pair > 90% of the visits were to the palatable colour after 8 days, which indicates that birds learned to discriminate the colours easily. This was important for testing social information transfer, but knowing exactly how different colour pairs were in spectrometer measurements was not relevant for our research question. We have now explained this in lines 123-126 and 416-424.

Line 536: “as a bound response variable” do you mean “bounded”?

Yes, we have now corrected this (line 487).

Lines 540-541: You don't have enough levels for this to make sense as a random effect. It's not reasonable to estimate variance from 3 options. It can only be a fixed effect; at least 5 or more levels are needed for a random effect.

We have now removed this random effect. We decided not to include the experiment as a fixed effect because we were not interested in differences among experiments and colour pairs were ambiguous and not biologically-informed. In addition, initial exploration of data suggested that learning was similar across all colour pairs (mentioned in lines 489-491).

Given that the two feeders (palatable, unpalatable) were swapped daily, the statistical analyses of feeder use should include time of day (or time after the switch). Early in day, the location is new

We switched the feeders daily to ensure that birds learned to associate taste with colour instead of location. We agree that this might influence birds' behaviour (i.e. they might take several visits to realise that the location has changed) but because our statistical analyses are already very complex, we decided not to include the time of the day in the models. However, this should only make our results more conservative (birds could not use multiple cues) and should not influence social information use which was the focus of the study.

Lines 554-555: There is not enough information here to understand how the networks were comprised. “the probability that two individuals were in the same group at a given time” isn't a complete explanation, because it would have a different value for the two nodes.

e.g., bird A is observed 10 times total and always in the presence of bird B, but bird B is observed 100 times, including 10 times with A, and 90 times without A
Edge A in presence of B is 1, but edge B in presence of A is 0.1

How was a single edge defined for the purpose of different network-based analyses here?

A network edge was calculated using the simple ratio index (SRI, e.g. see Farine & Whitehead 2015, Constructing, conducting, and interpreting animal social network analysis. J. Anim. Ecol. 84, 1144-1163) where the number of observations of the two birds foraging together was divided by the number of observations of them foraging together or either of the birds foraging alone. We have now clarified this and added the equation of SRI to the methods (lines 505-513).

Reviewer #2 (Remarks to the Author):

Review of “Social transmission alters the ecological dynamics of predators vs. prey”

This is a fascinating paper investigating the extent to which information about the palatability and

unpalatability of novel prey can be gained by observing the behaviour and choices of others. Social learning is a hot topic, and the work is significant because it attempts to elucidate the nature of information exchange in the context of food choice within and between two species under field conditions. I do however have the following comments which I hope will be of use in preparing any revision. In particular I suspect that more can be done to quantify the relative roles of asocial (direct) and social (indirect) learning, and more could be done to elucidate the (potentially complex) statistical properties of the "cumulative experience" models that are fitted.

1. The cornerstone of the paper is the fitted logistic model (equation 3 line 591 and subsequent extensions) based on the expected (through a_{ij}) distribution of observed feeding events by others and the actual feeding experience of the focal individual. I'm very reassured to see that simulations were run to evaluate the type I error rate and bias associated with the assumed model structure not only because of the difference between real and expected observations, but also because (i) the cumulative predictors ("N") in consecutive decisions are not independent (they all tend to increase over successive time intervals, see 2 below) and (ii) a response for individual i at time $< t$ is also part of a predictor for individual j at $> t$ (there is feedback) In addition to simulation, I therefore wonder whether it would be appropriate and possible to apply classical model diagnostics to ensure some key assumptions are not violated, notably independence of residuals and logit linearity.

We have followed the reviewer's suggestion and done this, and we have added the following section to the Supplementary information (lines 85-109):

"We examined the Pearson residuals from the model to assess the model assumptions, first plotting the residuals against the value of each predictor variable to assess whether the effects of each were linear on the log odds scale. Since the response variable is binary, this resulted in a banded pattern of residuals which is difficult to interpret. We therefore divided each predictor into a number of intervals and took the mean of residuals within each range allowing us to detect any trend in the residuals (see the R code 'GLMM models Orange Yellow final.r' in Supplementary data). In all cases we found that the average residual stayed constant across the range of each predictor variable, suggesting a linear relationship on the log-odds scales is at least a reasonable approximation for the effect of these variables.

Second, we analysed the residuals to test for autocorrelation in choice of palatable/unpalatable within each bird, i.e. the possibility that choices made close together in time were more likely to be the same than predicted by the model. We used a linear mixed effects model using the nlme package¹, with residual as the response variable, bird identity as a random effect, and choices within each bird correlated as a function of how close together they were in time (corCAR1 function, see the R code 'GLMM models Orange Yellow final.r'). This yielded an estimate of how strong the autocorrelation was as a function of time difference (ϕ parameter, correlation = time difference $^{\phi}$). We used the ϕ parameter to estimate how far apart in time choices needed to be to be considered independent (< 0.01 correlation), giving us a threshold of 15 secs (yellow/orange); 24 secs (blue/purple); 34 secs (red/green). We then cut out choices that were less than the

threshold time since the previous choice by that bird (yellow/orange 9%; blue/purple 9%; red/green 13%), and refitted the model. We saw some reduction in apparent statistical power, as expected by a smaller sample size, but the results were otherwise unchanged in each case, suggesting our findings were robust to inclusion/ exclusion of the autocorrelated data. We present the results of the reduced dataset in the main text, and used the reduced dataset for further analysis.”

When we undertook the second procedure, we also identified an error in our previous data filtering. To explain, we had previously attempted to exclude all records from the same bird, at the same feeder on consecutive seconds, since these were clearly part of the same feeding bout and not separate foraging decisions. Due to an oversight in our code, we had missed cases where two birds were feeding at the same time on different feeders. The second diagnostic procedure revealed these cases, and so we corrected our data prior to conducting the second analysis with the reduced dataset. This means our results have changed a little since the previous submission of our paper. The estimated effects are, overall, more consistent across diffusions, but statistical power is reduced for blue/purple and red/green (see the re-run simulations results). We report this here to explain why the results have changed despite the robustness to inclusion/exclusion of the autocorrelated data.

2. As noted above, the predictors are clearly multi-collinear: as time ticks by, all of the major component predictors will increase in size. I cannot find any note of this multi-collinearity (e.g. whether all other predictors in the model are simultaneously or sequentially controlled for in the Wald tests) or awareness of its implications (high multi-collinearity can potentially obscure otherwise strong relationships).

We were aware that collinearity was inevitable from the nature of the experiment, and indeed this was one of the two reasons why we conducted the simulations, to ensure that effects could be recovered and distinguished from one another despite the collinearity (the other was the approximation of using expected observations). We realise that we were too focused on explaining the latter and have added to Supplementary Information (lines 124-131) the following to our justification for the simulations:

“Furthermore, the nature of the experiment means that all the predictor variables are inevitably correlated since $N_{-,i}(t)$, $N_{+,i}(t)$, $O_{-,i}(t)$ and $O_{+,i}(t)$ will always increase over time. We were unable to drop or combine any predictors to reduce collinearity, since our aim was to estimate the social learning effects once asocial learning effects had been statistically controlled for. This means there is a potential risk for the effects of one variable to be obscured by the presence of another, or for effects of one variable to be misidentified as the effects of another variable. Consequently, we had two reasons to test whether our modelling approach can reliably detect and estimate the social learning effects of interest.”

The Wald tests used are the standard Wald statistics, which would detect significance after statistically controlling for all the other predictor variables in the model. Our simulations show that with the observed level of collinearity among the predictors we can still reliably detect the social

aversion learning effect, and that it tends to be conservatively estimated, showing that our finding of the social aversion learning effect can be trusted statistically. Conversely, the social appetitive learning tended to have inflated Type 1 error and overestimated effect sizes, which is likely a result of the collinearity- indeed we already included this section in the Supplementary information (lines 181-187):

“In contrast, there was an inflated type 1 error for a spurious effect of social appetitive learning ($p < 0.05$ and β_{soc+} estimated at < 0) of around 12.4% (see Table S6). The mean estimate of β_{soc+} also tended to be slightly more negative than its true value in the simulation with 95% C.I.s that also tended to over-estimate the effect (see Table S7). We suspect this occurred because opportunities to observe positive feeding events were relatively common (compared to negative feeding events), thus $O_{+,i}(t)$ was more highly correlated with $N_{+,i}(t)$, meaning a spurious effect of the former could be detected as a byproduct of asocial learning about the positive option.”

So we are suspicious of any apparent social appetitive learning effects, and have added the following amendment to the main text (lines 181-184):

“In the yellow/orange experiment there was some evidence that an increasing number of expected observations of conspecifics eating palatable almonds made birds more likely to choose the same palatable colour, but this effect did not follow the network and may be an artefact of the analysis (see Supplementary Information).”

3. I wondered about the mechanics of the binary choice and the role of time. Are consecutive choices of the same bird to a given feeder within a short time frame treated any differently from much more spread out (and clearly separate) visits (e.g. if green almonds are chosen three times in a row could it be because of spatial proximity, not learning). Also, it is ironic that seeing birds visit an unpalatable feeder serves to educate observers to avoid the same feeder. One might think choosing a particular coloured feeder would prompt others to follow rather than refrain (just as it does for palatable prey), which suggests that the observed disgust reaction is more important than the visit per se.

Indeed, it seems that there was some autocorrelation in choices that were close temporally, and consequently also spatially. We have made changes to the analysis to allow for this, see our response to point 1.

As the reviewer points out, our results suggest that observing a disgust response to unpalatable prey provides birds more salient social information compared to a positive feeding event (possibly because of risks of consuming toxic prey). We discuss this in lines 186-193.

4. Why was the individual station (it appears there were three of them) not entered as a predictor in the model? After all, it is possible that the birds apply location-specific rules initially e.g. avoid red almonds in station A and only learn later that it generalizes across the other stations. Likewise, although the effect of the demonstrators' identity (species and age class) is explored at length, were there any consistent patterns in the focal individual's identity in their propensity to learn from others (id nested within age class within species)?

Entering “station” as a predictor in the model without interactions with other predictors would only allow for the possibility that there are different biases towards/against unpalatable colour at each station. The only reason we can think that such differing biases would exist is that the birds that favoured each station, by chance, had different underlying biases, and this individual variation is already accounted for by the random effect. Even including interactions of “station” with the other predictors would only allow for the possibility of different learning rates at each station, which seems implausible and is not the effect that the reviewer describes.

The learning scenario described by the reviewer sounds plausible. To test for it we would have to split each predictor variable into three site-specific variables, and we could then compare a model in which the site-specific variables are used in place of the nonspecific variables, e.g. using AIC. However, we would be comparing a model where learning is site-specific with one in which it generalizes across sites: neither model would capture the scenario described, where location specific learning becomes generalized over time. We feel that a model that captures such a scenario would go beyond what we can extract from the data in this study, and is perhaps a goal for future work in which observations at each site are recorded directly.

As it stands, we cannot rule out the possibility that birds initially learn a site-specific rule that is generalized over time, but nor would such a scenario contradict the key findings of our study- that we have evidence of social avoidance learning in an open diffusion experiment.

We did also try fitting models in which we investigated the effects of learner’s age and species on the effects of learning, however, we found that this seemed to be stretching the data too thinly- we were unable to get the models to converge reliably (e.g. “Model is nearly unidentifiable: large eigenvalue ratio”, gradient 2 orders of magnitude away from tolerance for convergence), and did not consider the output meaningful.

5. The effect sizes in Table 1 seem remarkably small (with multiplicative effects of 0.99 very close to 1 despite significance) but I had to remind myself this was per (cumulative) choice. What is really needed here is an estimate of the size of asocial (direct) learning parameters so we can judge how important direct experience is compared to indirect experience. After all, if the argument in the paper is that social learning “alters ecological dynamics” (title) then it would help to know how much of a difference it makes compared to asocial learning alone.

We have added to Table 1 rows for the parameters estimating the strength of asocial learning “per feed”. Overall, the estimated effect “per expected observation” is a similar magnitude to the effect of direct experience (bear in mind that these are likely to be conservative estimates- see Supplementary Information). These also show the strong evidence for asocial learning referred to above.

6. I had a hard time understanding the meaning of “relative predation risk for unpalatable prey”. While I can guess, it needs a formal definition so we can understand why a value of 1 is consistent with no preference. Incidentally why does Fig 2A end at about 0.2 but it pick up again at 0.4 when the reversal learning experiment begins (perhaps some learning has already gone on in day 1)?

We agree that relative predation risk was confusing and have now changed the y axis to describe proportion of visits to the feeder which is easier to interpret. The difference between day 8 in Fig 2a and day 1 in Fig 2b is likely to be explained by rapid reversal learning in day 1, and we have now mentioned this in the results (lines 239-241).

7. The surprising result to me in Figure 3 is not that social learning goes on (e.g. more potential observations of others controlling for an individual's direct experience reduces the probability of a bad choice) but that asocial learning is not apparent at all (all the coloured lines overlapped). You have to be very careful here however. Note that there may be something rather special, for example, about the birds that have not visited unpalatable almonds themselves and yet have (potentially) observed 4-5 other birds doing so. See McElreath Statistical Rethinking 2nd edition for a discussion of the "selection-distortion effect" (aka "collider bias").

See our response to Reviewer 1 above. If we understand the reviewer correctly, selection-distortion effect is indeed in operation on the raw data, but when the random effect is included, we are able to detect the within bird patterns that indicate strong evidence of asocial learning.

8. The first experiment compared observed choice dynamics with models where the a_{ij} values were all identical (homogeneous). Why not also explore setting social learning coefficient to zero to help assess the importance of social learning (as when fitting the NBDA models in the reversal learning experiment)? More importantly, following the odd result outlined in (7) above, I think it would be informative to see whether we lose any predictive power by dropping asocial (direct) learning entirely and see if social learning explains most of the variability in the probability of choosing unpalatable almonds.

We have now compared AIC when dropping both the social effects and both the asocial effects as suggested. In addition, we dropped each social effect separately. We have added this in the methods (lines 560-564) and present the results in Table S6 in Supplementary Information. In all three experiments dropping the asocial effects had a huge effect on model fit, consistent with the new rows added to Table 1 for the asocial effects. In all cases dropping both social effects had, at least a meaningful effect on model fit (huge effect on yellow/orange). In the main yellow/orange experiment dropping either social effect had a large impact on AIC. In red/green and blue/purple, dropping the aversive social effect increases AIC by a small amount, whereas dropping the appetitive social effect decreased AIC, consistent again with the p values and confidence intervals in Table 1.

9. The experiments are great, and while I understand where it is heading, I wonder whether the implications are a little overdone or at least premature. For example do we really know that (line 1) "social transmission alters ecological dynamics" and (line 33) "accounting for social transmission is necessary to understand coevolutionary processes"? Clearly social learning can reduce the burden of predation and therefore help (line 271) "resolve a long-standing puzzle", but there are other excellent explanations for how aposematism spreads (e.g. kin selection, they survive encounters). The paper provides good field evidence of social learning about dietary preferences,

but it is hard on the basis of this experiment to really know how crucial it is to understanding signal evolution.

We agree that we cannot draw strong conclusions about the importance of social transmission on signal evolution. We have now toned this down in our conclusions (lines 300-305) and changed the title to "Social transmission in the wild reduces predation pressure on novel aposematic prey signals". We have also mentioned other explanations for aposematism in the introduction (lines 76-78).

My overall take is that this is an excellent study capitalizing on technological advances to monitor individual behaviour in well-established population. The paper is well written and argued and the results highlighting social learning in the context of diet choice are fascinating.

Thank you very much for the positive feedback!

Minor stuff

Line 95 its not really "negative social information", but information about something negative (unpalatability)

We have now changed this to "social information about positive and/or negative foraging experiences of others" (lines 101-102).

Line 113 no EVIDENCE OF species-level

Corrected.

Line 115 (and elsewhere) Perhaps better to say Day² (shorter and more explicit) than Day (polynomial)

We agree and have now used Day².

Line 189 reducing their likelihood OF choosing

Corrected.

Line 491 why end when > 50% (while it can go over 50% by chance, the expectation is 50%)

We agree and have now changed this to 50 %.

Lines 581, you probably need a "j not equal to i" here and elsewhere

We have now added this in lines 535 and 539.

Lines 667 et seq. Why bold fonts for O?

Thank you for pointing this out! We have now removed bold fonts.

Tom Sherratt

Reviewer #3 (Remarks to the Author):

This is a wonderfully written paper addressing an exciting topic. The authors have combined high-throughput tracking and cutting-edge social transmission analytical methods with an elegant experimental design that provides, in my mind, the most convincing demonstration yet of how social learning can promote the evolution of aggregated aposematic prey.

There is an inherent risk in studies such as this where inferences based on artificial paradigms (here, coloured almond flakes at mechanical bird feeders) are made about natural predator-prey interactions in the wild. In this case, I feel that the implications for how aposematic prey may evolve are justified.

Thank you for the positive comments!

One minor 'issue' is that we cannot actually tell whether the birds are learning about the colouration of the individual almond flakes or the colouration of the feeders in this study, since these are perfectly correlated. Nevertheless, as the authors state on line 198 onwards, this is exactly the kind of natural scenario you find with many aggregating lepidopteran and hemipteran aposematic prey. The only change I feel is needed in an overall excellent paper is to therefore briefly mention somewhere that the study's implications about aposematism are indeed restricted to aggregating rather than solitary prey.

Birds often ate the almond away from the feeder, so observers were likely to see others eating individual almond flakes. However, it is true that we cannot separate this from birds associating the colour with the feeders. We have now added to the conclusions that understanding how predators use social information about solitary prey requires further work (lines 303-305).

I find myself in the peculiar position of not being able to critique anything else about this paper, and I feel the authors should be congratulated for a wonderful piece of scholarship that has a good chance of becoming a classic in the field.

Thank you very much!!

REVIEWER COMMENTS

Reviewer #1 (Remarks to the Author):

The biggest issue with this study is that the manuscript misrepresents what is being measured. In the Title, Abstract, and Intro/Conclusion text, it's claimed that this study tests "how selection acts on prey defences" and how it acts on "aposematic prey signals". While this is an interesting study of social learning, it is not a measure of "predation pressure on ... aposematic prey signals", as the title claims. The granivorous behaviour measured here is not predation. A fixed feeder location that contains an infinite supply of homogeneous seed items is not prey, and does not provide a measure of predator-prey eco-evolutionary dynamics. Furthermore, within any given day of this experiment, an almond's phenotype was confounded with feeder location. How can this experiment allow us to separate learning a dichotomous location choice from learning aposematic signals per se? How does learning at a seed feeder measure selection on prey defences?

Example of problematic claims in first few paragraphs...

Lines 29-30 "social transmission among predators" "how selection acts on prey defences" "using artificial prey"

Line 32: show that... "predators learn about prey defences"

The sentence on Lines 33-34

Line 64 "predators vs. chemically defended prey"

Lines 65-66 "to test... whether this rapidly changing predator environment can alter the eco-evolutionary dynamics of prey defences"

This can be addressed by careful revisions to the claims about what is being measured in the Title, Abstract, and text.

Lines 78: Avian foraging on real arthropods, the most common aposematic prey group, is highly dynamic in space and time. e.g., Birds that forage on butterflies have to chase down individual prey. Insectivory in this form is not a social activity nor is there often an opportunity for one bird to see the specific traits of the prey caught or rejected by another individual. By contrast, in your experiment, the food items came from a specific location (and feeder location is confounded with almond phenotype). How is seed feeder use equivalent to selection via insectivory, or learning opportunities during real insectivory? I see no justification for presenting this paradigm as a measure of eco-evolutionary selection on prey.

I certainly agree with the authors that social learning research is relevant for understanding predator-prey interactions in general, e.g., lines 68-70. But homogenous seed items dispensed from a single location are not prey, and choice of a dichotomous seed feeder is not measuring the dynamics of predation.

Lines 103-104: It's not clear what you mean by 'models' and 'mimics' here, the explanation needs another sentence or two to justify this context. If all almonds are palatable during the experiment on Aug 3-13 Aug, then what are the almonds mimicking?

Lines 133-134: This is still unclear. I think what you mean to say is as follows: "If social avoidance learning occurs, then we expect birds to learn to avoid the unpalatable food option after having observed the experience of close social affiliates. We consequently devised a method to test this hypothesis by using network analysis to estimate the number of times an individual was expected to have viewed a social affiliate visit the unpalatable feeder. The association network was constructed..."

Line 155: "homogeneous" is not adequately explained, I can't tell what is being proposed here. Furthermore, the use of homogeneous here seems to be different from the way homogeneous is used on line 170-171. I think you are saying that social learning may occur but not follow the social network, but more clarification for the implications and diagnosis of this hypothesis is needed.

Line 158: same comment

Line 166: The use of "made" here is not correct. It can be fixed by changing the sentence to read: "After a greater number of expected observations of xxx, individuals were less likely to xxx..."

Line 179-181: Needs additional context from the vast literature on food preference learning in animals. Are avoidance reactions frequently more salient to learners? Is this result surprising?

Line 183-184: It's troubling that a result that is hard to explain is considered an artefact of the analysis, but others are not. Are you sure that "artefact of the analysis" is the best way to explain this variation?

Line 188: See comment above for line 179... there is a large experimental literature on mechanisms of food learning. This provides necessary background and context for your results & should be considered here.

Line 214: change "educated" to "informed"

Line 252: change "our social network quantifies" to "our social network approach estimates"

Line 287: "predator avoidance" is not being studied here at a dispensing feeder

Same comment for the sentence on lines 288-289, this sentence misrepresents what is actually being studied here.

Line 302-303: But does that occur often for animals that forage on arthropods, which are the most commonly defended prey? My personal experience studying insectivorous bird populations in North, Central and South America indicates that it does not.

Line 304: Not only were they aggregated, but food source location and signal traits were confounded. There is no evidence here that birds learned the specific traits of food taken by social groupmates,

because trait was confounded with location per each learning event.

Line 340: This figure is now much more useful now. However, in the caption, the phrase “was especially important” is not correct. What the figure shows is that social experience only explained choice variation among birds with low personal experience. “Especially” implies that x always explained variation in y, and especially so in the low category, but this is not what the figure shows. The figure shows that x does not explain variation in y except in the low category.

Additionally, my understanding is that the categories here are for visualization purposes only, and that in the analysis, personal experience is treated as a numerical variable. If that is correct, it should be explained to the reader in the caption.

Line 398: Add a statement explaining that these particular colours were chosen arbitrarily based on what was available for this particular dye product. You provided this information in the response to reviewers, but it is still missing from the manuscript. It should be added for the readers to understand the reasons underlying the design.

Line 420: change “visions” to “vision”

Reviewer #2 (Remarks to the Author):

I have re-reviewed the manuscript “Social transmission in the wild reduces predation pressure on novel aposematic prey signals” and think that the revisions have improved the paper considerably. My primary concerns were that: (1) in highlighting the role of social learning the role of direct (asocial) learning was being overlooked and (2) the fitted model was so complex and atypical (predictors are cumulative variables fed in part by the response), that more could be done to evaluate its properties (including whether further adjustments were necessary due to autocorrelation in the data).

It appears that direct asocial learning is indeed an important determinant of the odds of a bird choosing unpalatable almonds (Table 1 and Table S6) and that the former Figure 3 (which suggested that asocial learning was unimportant) was misleading due to a simple graphing error. While the focus of the paper is understandably on social learning, I think that knowing the influence and extent of asocial learning helps place the authors' results in context. Analyzing data based on expected observations will inevitably give challenges, especially in a social context, and I think the authors have done a good job in characterizing the logistic model they have fitted. I note that there was indeed evidence of autocorrelation (observations of a bird close in time are not independent) but the authors have dealt with it by exploring the effect of certain filtering thresholds (which I assume would not only influence the response variable, but also feed back into the predictors).

This is a fascinating but rather complicated paper, with a much to digest. On re-reading I have the following minor comments.

1) Table 1 is a vast improvement and is well set out. Since the slopes (beta coefficients in the logit model) are multiplied by the number of actual feeds on almonds (and/or the number of expected observations of others feeding), it would help to have an idea the mean total number of actual feeds and mean expected total number of observed feeds of others on palatable and unpalatable almonds. If nothing else it will provide context. Any odds ratio above 4 (or below -4) becomes pretty certain. So, in a well-connected big population then the beta coefficients for social learning might be less because the cumulative predictor values (number of observations of others) get so large.

2) The real (R) and expected (O) number of observed feeds are only equivalent if the social network is unchanged, a_{ij} is known with certainty, a_{ij} accurately reflects the opportunity to observe others, and there is no stochasticity. I appreciate that R and O are likely to be proportional (after all, the social model had some explanatory power) and that birds may feed away from the site they collected the almonds. Given the uncertainty however, it may be stretching too far to say that the beta coefficients are conservative estimates of the effect of an observation on a feeding decision (line 573).

3) In Table S1, I'm assuming that models with D(poly) also included D(linear) because it does not make much sense to have an interaction term without the associated main effects (it violates the principle of marginality).

Reviewer #3 (Remarks to the Author):

I think the authors have done a great job with addressing (minor) concerns regarding how this study relates to real-world systems. It seems from reading the other reviews that the title and abstract may have led some readers to believe the methods were based on more naturalistic scenarios involving real prey (I did not feel misled by this personally), but the authors have now thoroughly addressed this and the manuscript is much stronger (albeit perhaps now slightly understated from my perspective!).

As I stated in my previous review, and other reviewers have highlighted, whilst the design of this system does not allow us to determine exactly what the birds are learning in this study (spatial location of unpalatable prey, colour of feeder, colour of individual almonds etc), this is almost a moot point since it is the same scenario as we would find in nature with socially-aggregated, spatially restricted aposematic prey such as Hemipteran and Lepidopteran larva – the classic case studies for aposematism evolution. Determining what is being learned will make a great question for a future study, but I do not feel it is necessary here given the aims of this project.

Nevertheless, perhaps my enthusiasm got the better of me, and I appreciate the authors having changed the language in several places on exactly what we can infer about this study based on these results. I stand by my original point that this still provides some of the best evidence of how learning about

aposematism can transmit through a population of social predators, and I think the authors should be applauded on a well-executed and extremely exciting paper.

REVIEWER COMMENTS

Reviewer #1 (Remarks to the Author):

The biggest issue with this study is that the manuscript misrepresents what is being measured. In the Title, Abstract, and Intro/Conclusion text, it's claimed that this study tests "how selection acts on prey defences" and how it acts on "aposematic prey signals". While this is an interesting study of social learning, it is not a measure of "predation pressure on ... aposematic prey signals", as the title claims. The granivorous behaviour measured here is not predation. A fixed feeder location that contains an infinite supply of homogeneous seed items is not prey, and does not provide a measure of predator-prey eco-evolutionary dynamics. Furthermore, within any given day of this experiment, an almond's phenotype was confounded with feeder location. How can this experiment allow us to separate learning a dichotomous location choice from learning aposematic signals per se? How does learning at a seed feeder measure selection on prey defences?

Example of problematic claims in first few paragraphs...

Lines 29-30 "social transmission among predators" "how selection acts on prey defences" "using artificial prey"

Line 32: show that... "predators learn about prey defences"

The sentence on Lines 33-34

Line 64 "predators vs. chemically defended prey"

Lines 65-66 "to test... whether this rapidly changing predator environment can alter the eco-evolutionary dynamics of prey defences"

This can be addressed by careful revisions to the claims about what is being measured in the Title, Abstract, and text.

We agree that our experiment has some limitations (as any experiment!), and in addition to artificial prey approach, we would ideally study social transmission using both real predators and prey. However, conducting controlled field experiments with real predator-prey systems is often not feasible, and the use of artificial prey is a common method when studying predator-prey coevolution (e.g., Alatalo & Mappes 1996 Nature; Riipi et al. 2001 Nature). We have now clarified this in lines 99-101. Furthermore, our aim was to test how birds learn about novel prey signals, so we could not have used prey species that birds were already familiar with. We agree that this brings limitations to the interpretations of our results, and we have now toned down some of our conclusions. Specifically, we have removed "aposematic" from the title, toned down claims about eco-evolutionary dynamics (lines 64-66) and discussed learning about the signal vs. the feeder location (the side of the feeders was swapped daily, lines 307-310). However, we think that our study still provides important insights into predator-prey dynamics and we have decided to keep this predator-prey approach (supported by the encouraging comments from Reviewers 2 and 3).

Lines 78: Avian foraging on real arthropods, the most common aposematic prey group, is highly dynamic in space and time. e.g., Birds that forage on butterflies have to chase down individual prey. Insectivory in this form is not a social activity nor is there often an opportunity for one bird to see the specific traits of the prey caught or rejected by another individual. By contrast, in your

experiment, the food items came from a specific location (and feeder location is confounded with almond phenotype). How is seed feeder use equivalent to selection via insectivory, or learning opportunities during real insectivory? I see no justification for presenting this paradigm as a measure of eco-evolutionary selection on prey.

I certainly agree with the authors that social learning research is relevant for understanding predator-prey interactions in general, e.g., lines 68-70. But homogenous seed items dispensed from a single location are not prey, and choice of a dichotomous seed feeder is not measuring the dynamics of predation.

Please see our response to the comment above. Our experiment shows that birds use social information to make their foraging decisions. Contrary to previous experiments in captive conditions, our predators were wild and freely moving, which already added levels of complexity and “naturalness” to the experimental design. Investigating how birds learn about dispersed prey was not feasible in our study, and we used an experimental design that represented aggregated aposematic prey. There are many aposematic insects that are not mobile or live in large aggregations (e.g., many aphids, sawfly larvae, beetles and bugs). Furthermore, the concept of aposematism is not limited to animals only but many plants and plant parts (e.g., seeds) are aposematic (e.g., see Lev-Yadun, 2009, Plant-Environment Interactions). We have now clarified this in the introduction (lines 108-111). We do, however, agree that the use of aggregated prey brings some limitations to our conclusions, and more work is needed to understand how social information influences selection on solitary prey. We have discussed the relevance of prey aggregation in lines 234-240, and also highlight the need for further work to better understand the importance of social transmission in other predator-prey systems (lines 321-326).

Lines 103-104: It's not clear what you mean by 'models' and 'mimics' here, the explanation needs another sentence or two to justify this context. If all almonds are palatable during the experiment on Aug 3-13 Aug, then what are the almonds mimicking?

Before our reversal experiment, birds had acquired avoidance to unpalatable blue almonds (avoidance learning experiment 21-30 July). Palatable blue almonds in the reversal learning experiment were therefore mimicking these previously unpalatable almonds, and our aim was to test how quickly birds reversed their avoidance. This represents a situation where palatable mimics do not co-occur with their defended models (as mentioned in lines 286-287). We have now clarified this in the introduction: “This represents a situation where predators learn to avoid novel aposematic prey, and then encounter a population of palatable mimics that do not co-occur with their aposematic models” (lines 106-108).

Lines 133-134: This is still unclear. I think what you mean to say is as follows: “If social avoidance learning occurs, then we expect birds to learn to avoid the unpalatable food option after having observed the experience of close social affiliates. We consequently devised a method to test this hypothesis by using network analysis to estimate the number of times an individual was expected to have viewed a social affiliate visit the unpalatable feeder. The association network was constructed...”

Thank you for the suggestion! We have modified the sentences: “We next investigated whether birds used social information during avoidance learning. If social avoidance learning occurs, we would expect birds to learn to avoid unpalatable food after observing the negative experience of close social affiliates. We devised a method to test this hypothesis by using network analysis to estimate the number of times an individual was expected to have observed a social affiliate visiting the unpalatable feeder” (lines 138-142).

Line 155: “homogeneous” is not adequately explained, I can’t tell what is being proposed here. Furthermore, the use of homogeneous here seems to be different from the way homogeneous is used on line 170-171. I think you are saying that social learning may occur but not follow the social network, but more clarification for the implications and diagnosis of this hypothesis is needed.

We have now clarified this: “This means that social learning may occur, but it does not follow the association network (i.e. all birds having the same probability of observing each other), which would provide weaker evidence of social transmission” (lines 163-165). The use of homogeneous is the same in both sentences, and we have also clarified it in lines 180-181.

Line 158: same comment

We have clarified the previous definition (lines 163-165) and think that this sentence is now clear.

Line 166: The use of “made” here is not correct. It can be fixed by changing the sentence to read: “After a greater number of expected observations of xxx, individuals were less likely to xxx...”?

We have now corrected this as suggested.

Line 179-181: Needs additional context from the vast literature on food preference learning in animals. Are avoidance reactions frequently more salient to learners? Is this result surprising?

Although there is extensive literature on social learning of food choices, these studies have typically focused on social preference or avoidance learning only, and not compared these two information types. Furthermore, disentangling these two effects is often difficult, especially in simultaneous choice set-ups. We agree that this would be an interesting topic for further research, and have now added the following in lines 201-204: “This suggests that witnessing a strong response to unpalatable prey (e.g., vigorous beak wiping and head shaking) provides observers more salient social information than positive information about prey palatability, although this requires experimental tests that compare the effects of these two information types”.

Line 183-184: It’s troubling that a result that is hard to explain is considered an artefact of the analysis, but others are not. Are you sure that “artefact of the analysis” is the best way to explain this variation?

Our simulations to validate the modeling approach showed that the estimated effects of social avoidance learning were conservative, whereas the estimates of the effects of social appetitive learning (learning by observing positive feeding events) should only be inferred if we have strong

evidence of the effect following our social network (see Supplementary Information lines 181-190). Therefore, this specific result might be an artefact of the analysis (because we did not find the effect following the network), and the results of social avoidance learning are more reliable. We have now clarified this in lines 193-199: "While the simulations to validate our modeling approach showed that we can reliably detect social avoidance learning, we found that the estimates of the effects of social appetitive learning were less conservative (see Supplementary Information). This suggests that we should not make strong conclusions about learning from positive feeding events unless we have good evidence that this effect follows social network, and our result of social appetitive learning from conspecifics may therefore be an artefact of the analysis."

Line 188: See comment above for line 179... there is a large experimental literature on mechanisms of food learning. This provides necessary background and context for your results & should be considered here.

Please see our response to the previous comment. We have now mentioned that further work is needed to compare the effects of the two information types (lines 201-204).

Line 214: change "educated" to "informed"

Done.

Line 252: change "our social network quantifies" to "our social network approach estimates"

Done.

Line 287: "predator avoidance" is not being studied here at a dispensing feeder

Thank you for pointing that out. We have now changed this to "predators' avoidance" to clarify that we mean predators avoiding prey (line 303).

Same comment for the sentence on lines 288-289, this sentence misrepresents what is actually being studied here.

We think that this sentence is now clear, but have changed "defences" to "profitability" to be more specific (line 305).

Line 302-303: But does that occur often for animals that forage on arthropods, which are the most commonly defended prey? My personal experience studying insectivorous bird populations in North, Central and South America indicates that it does not.

As we state in lines 319-321, the importance of social information is likely to vary among predator and prey communities, but these differences are still poorly understood. For example, even if predators are not social foragers, they might have opportunities to gather social information from parents or other juveniles early in life. So far studies on avoidance learning of predators have focused on a few model species, and we have now highlighted the importance on further work to understand social transmission in other predator communities (lines 322-324): "Although this was the case in

our study, further work is needed to quantify social interactions in other predator communities, especially in areas rich in aposematic prey species (e.g., tropical environments²⁶)”.

Line 304: Not only were they aggregated, but food source location and signal traits were confounded. There is no evidence here that birds learned the specific traits of food taken by social groupmates, because trait was confounded with location per each learning event.

We agree that we cannot disentangle the specific cues that birds were using. However, we switched the side of the feeders daily to minimise learning about prey location. Birds also typically consumed the almonds away from the feeder, and observers were therefore more likely to learn about prey signals, rather than the location. Furthermore, birds improved their discrimination across the days of the experiment, which is unlikely if they were only learning about prey location (which changed every day). Previous experiments with captive birds also support the idea of birds using social information about prey signals (see references in the text). We have now addressed this in the discussion (lines 307-310).

Line 340: This figure is now much more useful now. However, in the caption, the phrase “was especially important” is not correct. What the figure shows is that social experience only explained choice variation among birds with low personal experience. “Especially” implies that x always explained variation in y, and especially so in the low category, but this is not what the figure shows. The figure shows that x does not explain variation in y except in the low category. Additionally, my understanding is that the categories here are for visualization purposes only, and that in the analysis, personal experience is treated as a numerical variable. If that is correct, it should be explained to the reader in the caption.

Thank you for pointing this out! We have now removed “especially” and clarified the caption as suggested.

Line 398: Add a statement explaining that these particular colours were chosen arbitrarily based on what was available for this particular dye product. You provided this information in the response to reviewers, but it is still missing from the manuscript. It should be added for the readers to understand the reasons underlying the design.

This information was included in the next paragraph of the methods: “For the other two learning experiments, we chose colour pairs that were available as food dye and as different from red and green in the visible spectrum as possible to avoid generalisation across experiments” (lines 435-437).

Line 420: change “visions” to “vision”

Done.

Reviewer #2 (Remarks to the Author):

I have re-reviewed the manuscript “Social transmission in the wild reduces predation pressure on novel aposematic prey signals” and think that the revisions have improved the paper considerably. My primary concerns were that: (1) in highlighting the role of social learning the role of direct (asocial) learning was being overlooked and (2) the fitted model was so complex and atypical (predictors are cumulative variables fed in part by the response), that more could be done to evaluate its properties (including whether further adjustments were necessary due to autocorrelation in the data).

It appears that direct asocial learning is indeed an important determinant of the odds of a bird choosing unpalatable almonds (Table 1 and Table S6) and that the former Figure 3 (which suggested that asocial learning was unimportant) was misleading due to a simple graphing error. While the focus of the paper is understandably on social learning, I think that knowing the influence and extent of asocial learning helps place the authors' results in context. Analyzing data based on expected observations will inevitably give challenges, especially in a social context, and I think the authors have done a good job in characterizing the logistic model they have fitted. I note that there was indeed evidence of autocorrelation (observations of a bird close in time are not independent) but the authors have dealt with it by exploring the effect of certain filtering thresholds (which I assume would not only influence the response variable, but also feed back into the predictors).

This is a fascinating but rather complicated paper, with a much to digest. On re-reading I have the following minor comments.

1) Table 1 is a vast improvement and is well set out. Since the slopes (beta coefficients in the logit model) are multiplied by the number of actual feeds on almonds (and/or the number of expected observations of others feeding), it would help to have an idea the mean total number of actual feeds and mean expected total number of observed feeds of others on palatable and unpalatable almonds. If nothing else it will provide context. Any odds ratio above 4 (or below -4) becomes pretty certain. So, in a well-connected big population then the beta coefficients for social learning might be less because the cumulative predictor values (number of observations of others) get so large.

Thank you for the suggestion. We have now included this information in Table 1.

2) The real (R) and expected (O) number of observed feeds are only equivalent if the social network is unchanged, a_{ij} is known with certainty, a_{ij} accurately reflects the opportunity to observe others, and there is no stochasticity. I appreciate that R and O are likely to be proportional (after all, the social model had some explanatory power) and that birds may feed away from the site they collected the almonds. Given the uncertainty however, it may be stretching too far to say that the beta coefficients are conservative estimates of the effect of an observation on a feeding decision (line 573).

We agree and have now changed this to “ Note that since $p_o \leq 1$, and $\beta_{soc-} = \beta_{soc-}' - p_o$, β_{soc-} is more likely to underestimate than overestimate the effect per observation of a negative feeding event” (lines 592-593).

3) In Table S1, I'm assuming that models with D(poly) also included D(linear) because it does not make much sense to have an interaction term without the associated main effects (it violates the principle of marginality).

Thank you for pointing that out. The models included also linear terms. We have now corrected this in the Tables S1 and S2.

Reviewer #3 (Remarks to the Author):

I think the authors have done a great job with addressing (minor) concerns regarding how this study relates to real-world systems. It seems from reading the other reviews that the title and abstract may have led some readers to believe the methods were based on more naturalistic scenarios involving real prey (I did not feel misled by this personally), but the authors have now thoroughly addressed this and the manuscript is much stronger (albeit perhaps now slightly understated from my perspective!).

As I stated in my previous review, and other reviewers have highlighted, whilst the design of this system does not allow us to determine exactly what the birds are learning in this study (spatial location of unpalatable prey, colour of feeder, colour of individual almonds etc), this is almost a moot point since it is the same scenario as we would find in nature with socially-aggregated, spatially restricted aposematic prey such as Hemipteran and Lepidopteran larva – the classic case studies for aposematism evolution. Determining what is being learned will make a great question for a future study, but I do not feel it is necessary here given the aims of this project.

Nevertheless, perhaps my enthusiasm got the better of me, and I appreciate the authors having changed the language in several places on exactly what we can infer about this study based on these results. I stand by my original point that this still provides some of the best evidence of how learning about aposematism can transmit through a population of social predators, and I think the authors should be applauded on a well-executed and extremely exciting paper.

Thank you for the very positive feedback!

REVIEWERS' COMMENTS

Reviewer #1 (Remarks to the Author):

The biggest issue with this study remains: the language in the main text still misrepresents what is being studied. This is a very interesting study of how information is socially transmitted during foraging in the wild. These findings surely have implications for our understanding of prey signals. However, the paradigm/measurement system used here is not a system of predators capturing prey. It would be straightforward to revise the language used throughout the main text to resolve this issue. I also note that you already describe the system accurately throughout the Methods, Figures and Tables, in contrast to the language used in the main text. I list the examples below, many of which were the same ones flagged in the last round.

The title still claims “reduces predation pressure” on “novel prey signals”

Lines 29-30 “here we show... that... social transmission among predators” “how selection acts on prey defences” “using artificial prey”

Line 32: we find that... “predators learn about prey defences”

The sentence on Lines 33-34, “this... reduces predation pressure from naïve predators”

Line 64 “we take advantage of... predators vs. chemically defended prey”

Lines 65-66 “to test how social transmission alters predators’ foraging behaviour, and whether this rapidly changing predator environment can alter predation pressure on novel prey signals”

Lines 97-98: “To determine if social transmission among predators shapes... defended prey...”

Line 100: “using artificial prey to test predator learning”

...Note that on lines 101-106, the experiment is accurately represented... the language there represents what is being measured!

Line 108: “artificial prey items (coloured almond flakes)”

Lines 162-170: “even if they were homogeneous” The writing in this section is still unclear. It’s not clear what “they” refers to... it seems to be “social effects”, but I can’t tell. Some careful revision of this paragraph is needed to help the reader understand what you are trying to say – it’s an important section.

Line 202: “strong response to unpalatable prey”

Line 203: "about prey palatability"

Line 205: "about prey palatability"

Line 229: "feeding on unpalatable prey"

Line 295: "sample previously unpalatable prey"

Line 298: "sample previously unpalatable prey"

Line 299: "accelerate predation"

Line 304: "naïve predators"

Line 308: "that birds use to discriminate different prey types"

Line 309: "prey location"

Lines 317-318: "In our study, social transmission rapidly altered the potential for selection to act on both chemically-defended prey and their mimics"

Line 319: "predators foraged"

Line 326: "we focused here on predator-prey interactions"

REVIEWERS' COMMENTS

Reviewer #1 (Remarks to the Author):

The biggest issue with this study remains: the language in the main text still misrepresents what is being studied. This is a very interesting study of how information is socially transmitted during foraging in the wild. These findings surely have implications for our understanding of prey signals. However, the paradigm/measurement system used here is not a system of predators capturing prey. It would be straightforward to revise the language used throughout the main text to resolve this issue. I also note that you already describe the system accurately throughout the Methods, Figures and Tables, in contrast to the language used in the main text. I list the examples below, many of which were the same ones flagged in the last round.

The title still claims “reduces predation pressure” on “novel prey signals”

Lines 29-30 “here we show... that... social transmission among predators” “how selection acts on prey defences” “using artificial prey”

Line 32: we find that... “predators learn about prey defences”

The sentence on Lines 33-34, “this... reduces predation pressure from naïve predators”

Line 64 “we take advantage of... predators vs. chemically defended prey”

Lines 65-66 “to test how social transmission alters predators’ foraging behaviour, and whether this rapidly changing predator environment can alter predation pressure on novel prey signals”

Lines 97-98: “To determine if social transmission among predators shapes... defended prey...”

Line 100: “using artificial prey to test predator learning”

...Note that on lines 101-106, the experiment is accurately represented... the language there represents what is being measured!

Line 108: “artificial prey items (coloured almond flakes)”

We respectfully disagree and have decided not to change the wording for the reasons described above.

Lines 162-170: “even if they were homogeneous” The writing in this section is still unclear. It’s not clear what “they” refers to... it seems to be “social effects”, but I can’t tell. Some careful revision is of this paragraph is needed to help the reader understand what you are trying to say – it’s an important section.

We have changed “they” to “the effects” to be clearer (line 166).

Line 202: “strong response to unpalatable prey”

Line 203: “about prey palatability”

Line 205: “about prey palatability”

Line 229: “feeding on unpalatable prey”

Line 295: “sample previously unpalatable prey”

Line 298: “sample previously unpalatable prey”

Line 299: “accelerate predation”

Line 304: “naïve predators”

Line 308: “that birds use to discriminate different prey types”

Line 309: “prey location”

Lines 317-318: “In our study, social transmission rapidly altered the potential for selection to act on both chemically-defended prey and their mimics”

Line 319: “predators foraged”

Line 326: “we focused here on predator-prey interactions”

We respectfully disagree and have decided not to change the wording for the reasons described above.